# Advancing Constrained Monotonic Neural Networks: Achieving Universal Approximation Beyond Bounded Activations

Davide Sartor [* 1]  Alberto Sinigaglia [* 2]  Gian Antonio Susto [1 2]

## Abstract

Conventional techniques for imposing monotonicity in MLPs by construction involve the use of non-negative weight constraints and bounded activation functions, which pose well-known optimization challenges. In this work, we generalize previous theoretical results, showing that MLPs with non-negative weight constraint and activations that saturate on alternating sides are universal approximators for monotonic functions. Additionally, we show an equivalence between the saturation side in the activations and the sign of the weight constraint. This connection allows us to prove that MLPs with convex monotone activations and non-positive constrained weights also qualify as universal approximators, in contrast to their non-negative constrained counterparts. Our results provide theoretical grounding to the empirical effectiveness observed in previous works while leading to possible architectural simplification. Moreover, to further alleviate the optimization difficulties, we propose an alternative formulation that allows the network to adjust its activations according to the sign of the weights. This eliminates the requirement for weight reparameterization, easing initialization and improving training stability. Experimental evaluation reinforces the validity of the theoretical results, showing that our novel approach compares favourably to traditional monotonic architectures.

*Equal contribution [1]Department of Information Engineering, University of Padova, Padova (PD), Italy [2]Human Inspired Technology Research Centre, University of Padova, Padova (PD), Italy. Correspondence to: Davide Sartor <davide.sartor.4@phd.unipd.it>, Alberto Sinigaglia <alberto.sinigaglia@phd.unipd.it>.

*Proceedings of the $42^{nd}$ International Conference on Machine Learning*, Vancouver, Canada. PMLR 267, 2025. Copyright 2025 by the author(s).

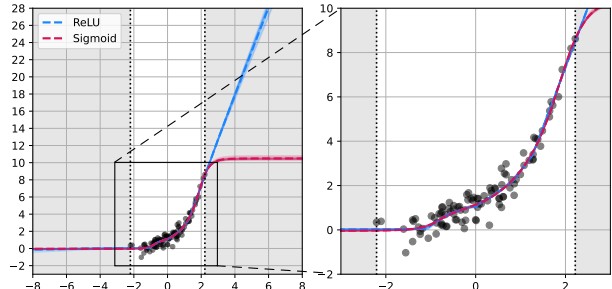

Figure 1. Monotone MLPs with weight-constraint and bounded activations (pink) and our proposed approach based on ReLU (blue). The former can only represent bounded functions and, thus, cannot extrapolate the data trend, which is important in many domains, such as time-series analysis and predictive maintenance.

## 1. Introduction

Monotonic neural networks represent a pivotal shift in deep learning. They bridge the gap between high-capacity non-linear models and the need for interpretable, consistent outputs in various applications. Monotonic MLPs preserve monotonic input-output relationships, making them particularly suitable for domains that require justified and transparent decisions (Gupta et al., 2016; Nguyen & Martínez, 2019). Furthermore, monotonic MLPs have been exploited to build novel architectures for density estimators (Chilinski & Silva, 2020; Omi et al., 2019; Tagasovska & Lopez-Paz, 2019), survival analysis (Jeanselme et al., 2023), and remaining useful life (Sánchez et al., 2023). In general, the enforcement of constraints on the model architecture guarantees certain desired properties, such as fairness or robustness. Furthermore, explicitly designing the model with inductive biases that exploit prior knowledge has been shown to be fundamental for efficient generalization (Dugas et al., 2000; Milani Fard et al., 2016; You et al., 2017). For this reason, the use of monotonic networks can help both in improving performance (Mitchell, 1980) and in data efficiency (Veličković, 2019).

Recent works in this field usually fall into one of the following two categories: 'soft monotonicity' and 'hard monotonicity'. Soft monotonicity employs optimization constraints (Gupta et al., 2019; Sill & Abu-Mostafa, 1996), usually as additional penalty terms in the loss. This class of approaches benefits from its simple implementation and inexpensive computation. They exploit the power of Multi-Layer Perceptrons (MLPs) to be able to approximate arbitrary functions. Since penalties are usually applied to dataset samples, monotonicity is enforced only *in-distribution*, therefore struggling to generalize the constraint out-of-distribution.

Hard monotonicity instead imposes constraints on the model architecture to ensure monotonicity by design (Wehenkel & Louppe, 2019; Nolte et al., 2023). The simplest way to do so is to constrain the MLP weights to be non-negative and to use monotonic activations (Daniels & Velikova, 2010). The methods proposed in the literature that exploit this parametrization (Daniels & Velikova, 2010; Wehenkel & Louppe, 2019) require the usage of bounded activations, such as sigmoid and hyperbolic tangent, which introduce well-known optimization challenges due to vanishing gradients (Dubey et al., 2022; Ravikumar & Sriraman, 2023; Szandała, 2021; Nair & Hinton, 2010; Glorot & Bengio, 2010; Goodfellow et al., 2013). This shortcoming is even more evident in monotonic NNs with non-negative weights, where bounded activations make the initialization even more crucial for optimization. As discussed in Appendix A.2, poor initialization can lead to saturated activations at the beginning of training, thus significantly slowing it down. Furthermore, the use of bounded activations leads to MLPs that can only represent bounded functions, which may hinder generalization, as shown in Figure 1.

Indeed, most recent advances in NNs use activations in the family of rectified linear functions, such as the popular ReLU activation (Vaswani, 2017; He et al., 2016). However, the use of ReLU activations in MLPs with non-negative weights is problematic. In fact, an MLP that uses convex activations (such as ReLU) in conjunction with non-negative weights can only approximate convex functions, which severely limits applications (Daniels & Velikova, 2010; Mikulincer & Reichman, 2022). For this reason, many approaches in the literature still rely on including bounded activations in the architecture in order to ensure universal approximation abilities of the network.

The primary aim of this work is to extend the theoretical basis of monotonic-constrained MLPs, by showing that it is still possible to achieve universal approximation using activations that saturate on one side, such as ReLU. To show that these new findings are not just theoretical tools, we create a new architecture that only uses saturating activations with performances comparable to state-of-the-art. Our contributions can be summarized as follows:

1. We show that constrained MLPs that alternate left-saturating and right-saturating monotonic activations can approximate any monotonic function. We also demonstrate that this can be achieved with a constant number of layers, which matches the best-known bound for threshold-activated networks.

2. Contrary to the non-negative-constrained formulation, we prove that an MLP with at least 4 layers, non-positive-constrained weights, and ReLU activation is a universal approximator. More generally, this holds true for any saturating monotonic activation.

3. We propose a simple parametrization scheme for monotonic MLPs that (i) can be used with saturating activations; (ii) does not require constrained parameters, thus making the optimization more stable and less sensitive to initialization; (iii) does not require multiple activations; (iv) does not require any prior choice of alternation of any activation and its point reflection.

Our discussion will primarily focus on ReLU activations, which are widely used in the latest advancements in deep learning. However, the results apply to the broader family of monotonic activations that saturate on at least one side. This includes most members of the family of ReLU-like activations such as exponential, ELU (Clevert et al., 2016), SeLU (Klambauer et al., 2017), CELU (Barron, 2017), SReLU (Jin et al., 2016), and many more.

## 2. Related work

Monotonicity in neural network architectures is an active area of research that has been addressed both theoretically and practically. Prior work can be broadly classified into two categories: architectures designed with built-in constraints (hard monotonicity) and those employing regularization and heuristic techniques to enforce monotonicity (soft monotonicity). Our contribution falls into the former category.

### 2.1. Hard Monotonicity

Hard-monotonicity aims at building MLPs with provably monotonicity for any point of the input space. They do so by constructing the MLP so that only monotonic functions can be learned. Initial attempts were Deep Lattice Networks (You et al., 2017) and methods constraining all weights to have the same sign exemplify this approach (Dugas et al., 2009; Runje & Shankaranarayana, 2023; Kim & Lee, 2024). However, constraining the parameters to be non-negative violates the original MLP formulation, thus invalidating the universal approximation theorem. Indeed, the universal ap-

Code available at github.com/AMCO-UniPD/monotonic.

proximation capability of the architecture is proven only under the condition that the threshold function is used as activation and that the network is at least 4 layers deep (Runje & Shankaranarayana, 2023). Furthermore, the non-negative parameter constraint also creates issues from an initialization standpoint, as the assumption for popular initializers might be violated for positive semidefinite matrixes. Only recently, new architectures have been proposed with novel techniques: adapting the Deep Lattice framework to MLPs (Yanagisawa et al., 2022), working with multiple activations (Runje & Shankaranarayana, 2023), and constraining the Lipschitz constant (Raghu et al., 2017). However, Runje & Shankaranarayana (2023) requires the usage of multiple activations, and an a priori split of the layer neurons between them, which might be suboptimal or require additional tuning, and Nolte et al. (2023) relies on very specific activations to control such property, as reported by the authors. In contrast, our work aims to overcome these drawbacks by improving flexibility without compromising the monotonicity constraint.

## 2.2. Soft Monotonicity

Soft-monotonicity aims to build monotonic MLPs by modifying the training instead of the architecture, either using heuristics or regularizations. Techniques such as point-wise penalty for negative gradients (Gupta et al., 2019; Sill & Abu-Mostafa, 1996) and the use of Mixed Integer Linear Programming (MILP) for certification (Liu et al., 2020) have been proposed. These methods maintain considerable expressive power but do not guarantee monotonicity. Additionally, the computational expense required for certifications, such as those using MILP or Satisfiability Modulo Theories (SMT) solvers, can be prohibitively high.

## 3. Monotone MLP

A function $f : \mathbb{R}^d \to \mathbb{R}$ is said to be monotone non-decreasing with respect to $x_i$, if given $x_i^0, x_i^1 \in \mathbb{R}, i \in [1, d]$, has the following property:

$$x_i^0 \leq x_i^1 \Rightarrow f(x_1, \ldots, x_i^0, \ldots, x_d) \leq f(x_1, \ldots, x_i^1, \ldots, x_d) \tag{1}$$

And similarly, a function $f : \mathbb{R}^d \to \mathbb{R}$ is said to be monotone non-increasing with respect to $x_i$ if:

$$x_i^0 \leq x_i^1 \Rightarrow f(x_1, \ldots, x_i^0, \ldots, x_d) \geq f(x_1, \ldots, x_i^1, \ldots, x_d). \tag{2}$$

*Remark* 3.1. Given $f(x), g(x)$ monotonic non-decreasing, and $h(x), u(x)$ monotonic non-increasing, $f \circ g$ is monotonic non-decreasing, $f \circ h$ is monotonic non-increasing, and $u \circ h$ is monotonic non-decreasing.

In this work, we will only focus on parametrizing non-decreasing functions, as the monotonicity can be reversed

by simply inverting the sign of the inputs. [1]

An MLP is defined as a parametrized function $f_\theta$ obtained as composition of alternating affine transformations $l_\theta^{(i)}$ and non-linear activations $\sigma_\theta^{(i)}$:

$$f_\theta(x) = l_\theta^{(1)} \circ \sigma_\theta^{(1)} \ldots \sigma_\theta^{(n-1)} \circ l_\theta^{(n)}. \tag{3}$$

A straightforward approach to building a provable monotonic MLP that respects Equation (1) is to impose constraints on its weights and activations, forcing each term in Equation (3) to be monotonic. For affine transformations $l_\theta^{(i)}(x) = W^{(i)}x + b^{(i)}$, we only need to enforce the Jacobian to be non-negative, which is simply the matrix $W^{(i)}$, while monotonic activations are, by definition, monotonic.

To optimize the resulting MLP with unconstrained gradient-based approaches, the non-negative weight constraint is obtained using reparametrization, i.e. $l_\theta^{(i)}(x) = g(W^{(i)})x + b^{(i)}$ for some differentiable $g : \mathbb{R} \to \mathbb{R}_+$. Typical reparametrizations use absolute value or squaring.

### 3.1. Known universal approximation conditions

Despite their surprising performance, one critical flaw of existing MLP architectures based on weight constraints is the narrow choice of activation functions. Constrained MLPs have been shown to be universal function approximators for monotonic functions, provided the activation is chosen to be the threshold function, and the number of hidden layers is greater than the dimension of the input variable (Daniels & Velikova, 2010). Just recently, this result has been drastically improved to a constant bound, proving that four layers are sufficient to have universal approximation properties (Mikulincer & Reichman, 2022), when using the threshold function as activation. Therefore, practical implementations still resort to bounded activations such as sigmoid, tanh, or ReLU6.

On the other hand, the use of convex activations like ReLU in a constrained MLP severely limits the expressivity of the network. To understand why this is the case, consider that:

**Proposition 3.2.** *The composition of monotonic convex functions is itself monotonic convex.*

Since affine transformations are simultaneously convex and concave, if the activation is chosen to be a monotonic convex function, then the constrained MLP will only be able to approximate monotonic convex functions. Despite this clear disadvantage, there is interest in ReLU activated monotonic MLPs due to their properties and their performances shown in the unconstrained case (Glorot & Bengio, 2010; Hein et al., 2019). Runje & Shankaranarayana (2023) propose a way to introduce ReLU activation in the

---

[1]In the rest of the paper we will use "monotonic" as shorthand for "monotonic non-decreasing", unless otherwise specified

network. The architecture uses multiple activation functions derived from a primitive activation $\sigma(x)$, its point reflection $\sigma'(x) = -\sigma(-x)$, and, in particular, a bounded sigmoid-like activation. While the ReLU activations are ignored in the subsequent theoretical analysis, this last bounded activation function is used to ensure the universal approximation property.

An upper-bound on the minimum required number of layers to ensure universal approximation of their architecture is derived from the result in Daniels & Velikova (2010). Although this bound scales linearly with the number of input dimensions, the authors observe good empirical performance with just a few layers. In this work, we extend the result of (Mikulincer & Reichman, 2022), showing that a constant number of layers is indeed sufficient, even when using non-threshold activations. This result explains the empirical observations of Runje & Shankaranarayana (2023), and suggests that one of the three activations employed might not be necessary.

### 3.2. Universal approximation theorem for non-threshold activations

From Proposition 3.2, it follows that a constrained MLP with ReLU activations cannot be a universal approximator for monotone functions. However, two MLP layers can approximate the Heaviside function arbitrarily well if the activations alternate between ReLU and its point reflection, as shown in Figure 2. From this observation, it is possible to use the result of (Daniels & Velikova, 2010) directly, showing that alternating activations between ReLU and its point reflection is sufficient to construct universal monotonic approximators. However, the resulting bound on the required number of layers is linear in the input dimension.

In Appendix A.1 we use similar reasoning and the results of (Mikulincer & Reichman, 2022) to achieve a loose but constant bound of 8 layers. However, in this section, we will derive a tighter bound that instead matches and generalizes

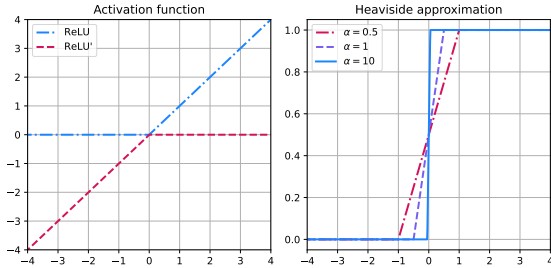

*Figure 2.* Constructions of Heaviside function using a composition of ReLU and its point reflection ReLU' to obtain $\text{ReLU}(\text{ReLU}'(\alpha x - 0.5) + 1) = \text{ReLU}'(\text{ReLU}(\alpha x + 0.5) - 1)$

the result derived in Mikulincer & Reichman (2022), while applying to a broader class of activation functions. Specifically, we will prove that 4 layers are sufficient, provided that the activations alternate saturation sides.

**Definition 3.3.** Given a function $\sigma : \mathbb{R} \to \mathbb{R}$ saturates right/left if the corresponding limit exists and is finite. That is, $\sigma$ is right-saturating if $\sigma(+\infty) := \lim_{x \to +\infty} \sigma(x) \in \mathbb{R}$, and it is left-saturating if $\sigma(-\infty) := \lim_{x \to -\infty} \sigma(x) \in \mathbb{R}$. We will denote the set of right-saturating activations as $\mathcal{S}^+$ and the set of left-saturating activations as $\mathcal{S}^-$.

**Proposition 3.4.** *For every MLP with non-negative weights and activation $\sigma(x)$, and for any $a \in \mathbb{R}_+, b \in \mathbb{R}$, there exists an equivalent MLP with non-negative weights and activation $a\sigma(x) + b$.*

In the following proofs, thanks to Proposition 3.4, we will only consider activations that saturate to zero.

The main result we will prove is the following.

**Theorem 3.5.** *An MLP $g_\theta : \mathbb{R}^d \to \mathbb{R}$ with non-negative weights and 3 hidden layers can interpolate any monotonic non-decreasing function $f(x)$ on any set of $n$ points, provided that the activation functions are monotonic non-decreasing and alternate saturation sides. That is, in addition to monotonicity, either of the following holds:*

$$\sigma^{(1)} \in \mathcal{S}^-, \sigma^{(2)} \in \mathcal{S}^+, \sigma^{(3)} \in \mathcal{S}^- \tag{4}$$

$$\sigma^{(1)} \in \mathcal{S}^+, \sigma^{(2)} \in \mathcal{S}^-, \sigma^{(3)} \in \mathcal{S}^+ \tag{5}$$

The first step is proving that hidden units in the first layer can approximate piecewise-constant functions on specific half-spaces.

**Lemma 3.6.** *Consider an arbitrary hyperplane defined by $\alpha^T (x - \beta) = 0$, $\alpha \in \mathbb{R}_+^d$ and $\beta \in \mathbb{R}^d$, and the open half-spaces $A^+ = \{x : \alpha^T (x - \beta) > 0\}$, $A^- = \{x : \alpha^T (x - \beta) < 0\}$. The i-th neuron in the first hidden layer of an MLP with non-negative weights can approximate* [2]:

$$h_i^{(1)}(x) \approx \begin{cases} \sigma^{(1)}(+\infty), & \text{if } x \in A^+ \\ \sigma^{(1)}(-\infty), & \text{if } x \in A^- \\ \sigma^{(1)}(0), & \text{otherwise} \end{cases}$$

*Proof.* Denote by $w$ the weights and by $b$ the bias associated with the hidden unit under consideration. Then, setting the parameters to $w = \lambda\alpha^T$ and $b = \lambda\alpha^T\beta$ and taking the limit we have that:

$$h_i^{(1)}(x) \approx \lim_{\lambda \to +\infty} \sigma^{(1)}(wx + b) = \lim_{\lambda \to +\infty} \sigma^{(1)}\left(\lambda\alpha^T(x - \beta)\right)$$

The limit is either $\sigma^{(1)}(+\infty)$, $\sigma^{(1)}(-\infty)$ or $\sigma^{(1)}(0)$ depending

---

[2]Note that $\sigma^{(1)}(\pm\infty)$ need not be finite.

on the sign of $\alpha^T (x - \beta)$, proving that:

$$h_i^{(1)}(x) \approx \begin{cases} \sigma^{(1)}(+\infty), & \text{if } \alpha^T (x - \beta) > 0 \\ \sigma^{(1)}(-\infty), & \text{if } \alpha^T (x - \beta) < 0 \\ \sigma^{(1)}(0), & \text{if } \alpha^T (x - \beta) = 0 \end{cases}$$

$\square$

The second step is to prove that one hidden layer can perform intersections of subspaces under specific conditions. In our construction, these will be either half-spaces or intersections of specific half-spaces.

**Lemma 3.7.** *Consider the intersection $A = \bigcap_{i=0}^{n} A_i$, for $A_1, \ldots, A_n$ subsets of $\mathbb{R}^d$. For any $\gamma$ in the image of $\sigma^{(k)}$, a single unit $h_j^{(k)}$ in the $k$-th hidden layer of an MLP with non-negative weights can approximate:*

$$h_j^{(k)}(x) \approx \gamma \mathbb{1}_A(x)$$

*provided that $h_i^{(k-1)}(x) \approx 0$ for $x \in A_i$, and either:*

- $\sigma^{(k)} \in \mathcal{S}^-$ and $h_i^{(k-1)}(x) < 0$ for $x \notin A_i$
- $\sigma^{(k)} \in \mathcal{S}^+$ and $h_i^{(k-1)}(x) > 0$ for $x \notin A_i$

*Proof.* Denote by $w$ the weights and by $b$ the bias associated with the hidden unit under consideration. Then, setting the weights to $w = \lambda \mathbf{1}^T$ and taking the limit we have that:

$$h_j^{(k)}(x) \approx \lim_{\lambda \to +\infty} \sigma^{(k)} \left( w h^{(k-1)}(x) + b \right)$$
$$= \lim_{\lambda \to +\infty} \sigma^{(k)} \left( b + \lambda \sum_{i=0}^{n} h_i^{(k-1)}(x) \right)$$

Note that in any case, if $x \in \bigcap_{i=1}^{n} A_i$, then $\lambda \sum_i h_i^{(k-1)}(x) \approx 0$, and the limit simply reduces to $\sigma^{(k)}(b)$. On the other hand, for $x \notin \bigcap_{i=1}^{n} A_i$, the limit can be either $\sigma^{(k)}(\pm\infty)$ depending on the sign of $h_i^{(k-1)}(x)$. When $\sigma^{(k)} \in \mathcal{S}^-$ and $h_i^{(k-1)}(x) < 0$, the limit is simply $\sigma^{(k)}(-\infty) = 0$. Similarly, when $\sigma^{(k)} \in \mathcal{S}^+$ and $h_i^{(k-1)}(x) > 0$ the limit is $\sigma^{(k)}(+\infty) = 0$.

In both cases, for any $\gamma$ in the image of $\sigma^{(k)}$ we can find a bias value $b$ so that:

$$h_j^{(k)}(x) \approx \gamma \mathbb{1}_A(x) = \begin{cases} \sigma^{(k)}(b) = \gamma, & \text{if } x \in \bigcap_{i=1}^{n} A_i, \\ \sigma^{(k)}(\pm\infty) = 0, & \text{otherwise} \end{cases}$$

$\square$

Thanks to Lemma 3.6 and Lemma 3.7, we can now prove the main result, that is Theorem 3.5. We will only prove the case $\sigma^{(1)} \in \mathcal{S}^-$, $\sigma^{(2)} \in \mathcal{S}^+$, $\sigma^{(3)} \in \mathcal{S}^-$. The proof for the opposite case follows the same structure and is reported in Appendix A.4.

*Proof of Theorem 3.5.* Assume, without loss of generality, that the points $x_1, \ldots, x_n$ are ordered so that $i < j \implies f(x_i) \leq f(x_j)$, with ties resolved arbitrarly. We will proceed by construction, layer by layer.

**Layer 1** Since the function to interpolate is monotonic, for any couple of points $x_i, x_j$ with $i < j$, it is possible to find a hyperplane defined by $\alpha_{j/i}^T (x - \beta_{j/i}) = 0$ for some $\beta_{j/i} \in \mathbb{R}^d$ and some non-negative normal $\alpha_{j/i} \in \mathbb{R}_+^d$, such that $x_j \in A_{j/i}^+, x_i \in A_{j/i}^-$, where $A_{j/i}^+$ and $A_{j/i}^-$ denote the positive and negative half spaces respectively.

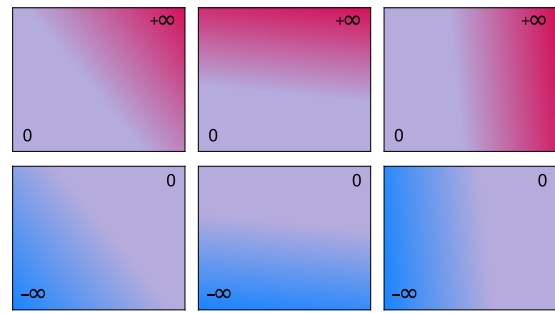

*Figure 3.* Example of representable functions at layer 1.

Using Lemma 3.6, we can ensure that it is possible to have:

$$\begin{cases} h_i^{(1)}(x) \approx \sigma^{(1)}(-\infty) = 0, & \text{if } x \in A_{j/i}^- \; \forall i, j \\ h_i^{(1)}(x) \approx \sigma^{(1)}(+\infty) > 0, & \text{otherwise} \end{cases} \quad (6)$$

A visual example is shown in Figure 3.

**Layer 2** Let us construct the set $A_i^{(2)} = \bigcap_{j:j>i} A_{j/i}^-$. Note that the sets $A_i^{(2)}$ always contain $x_i$ and do not contain any $x_j$ for $j > i$. Using Equation (6), we can apply Lemma 3.7, which ensures that it is possible to have the following[3]:

$$\begin{cases} h_i^{(2)}(x) \approx 0, & \text{if } x \notin A_i^{(2)} \\ h_i^{(2)}(x) \approx \gamma^{(2)} < 0, & \text{otherwise} \end{cases} \quad (7)$$

A visual example is shown in Figure 4.

**Layer 3** Consider $A_i^{(3)} = \bigcap_{j:j<i} \bar{A}_j^{(2)}$, where $\bar{A}_j^{(2)}$ is the complement of $A_j^{(2)}$. Using Equation (7) we can once again apply Lemma 3.7, which ensures that it is possible to have the following[4]:

$$h_i^{(3)}(x) \approx \gamma^{(3)} \mathbb{1}_{A_i^{(3)}}(x) \quad (8)$$

Now, we will show that $A_i^{(3)}$ represents a level set, i.e. $x_j \in A_i^{(3)} \iff f(x_j) \geq f(x_i)$. To do so, consider that

---

[3] In this case $\gamma^{(2)} < 0$ since by assumption $\sigma^{(2)}$ saturates right.
[4] In this case $\gamma^{(3)} > 0$ since by assumption $\sigma^{(3)}$ saturates left.

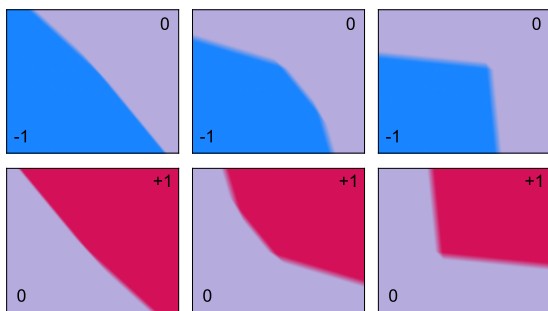

Figure 4. Example of representable functions at layer 2.

$\bar{A}_i^{(3)} = \bigcup_{j:j<i} A_j^{(2)}$. Since $x_j \in A_j^{(2)}$, then $x_j \in \bar{A}_i^{(3)}$ for $j < i$. Similarly since $x_j$ is the largest point contained in $A_j^{(2)}$, $\bar{A}_i^{(3)}$ cannot contain $x_i$ or any point larger than $x_i$. This shows that $A_i^{(3)}$ contains exactly the points $\{x_j : f(x_j) \geq f(x_i)\}$. A visual example is shown in Figure 5.

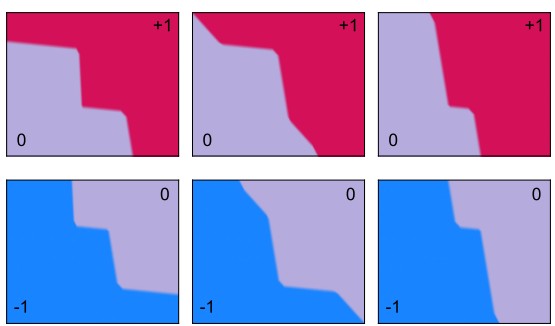

Figure 5. Example of representable functions at layer 3.

**Layer 4**    To conclude the proof, simply take the weights at the fourth layer to be :

$$w = \left[ \frac{f(x_1) - b}{\gamma^{(3)}}, \frac{f(x_2) - f(x_1)}{\gamma^{(3)}}, \ldots, \frac{f(x_n) - f(x_{n-1})}{\gamma^{(3)}} \right]$$

Since the points are ordered, this ensures that $w$ contains all non-negative terms, when bias term $b$ is taken to be $b \leq f(x_1)$. Defining $f(x_0) = b$, the output of the MLP can be expressed as:

$$g_\theta(x) = w^T h^{(3)}(x) + b = b + \sum_{j=1}^{n} (f(x_j) - f(x_{j-1})) \, \mathbb{1}_{A_j^{(3)}}(x)$$
(9)

Evaluating Equation (9) at any of the points $x_i$, it reduces to the telescopic sum:

$$g_\theta(x_i) = f(x_1) + \sum_{j=2}^{i} (f(x_j) - f(x_{j-1})) = f(x_i) \quad (10)$$

Thus proving that the network correctly interpolates the target function.    □

### 3.3. Non-positive constrained monotonic MLP

Consider the simple modification of the standard constrained MLP approach described in Equation (3). However, instead of constraining the weights to be non-negative, they are constrained to be non-positive. Although this simple modification might seem inconsequential, we will show that this is not the case. Indeed, we will show that a non-positive constrained MLP satisfies the conditions of Theorem 3.5, as long as the activation function saturates on at least one side. This includes convex activations like ReLU, which provably do not yield universal approximators in the non-negative constrained weight setting. Note that by Remark 3.1, an MLP defined according to Equation (3) is still monotone for an even number of non-positively constrained layers; therefore, it is still possible to construct provably monotonic networks using non-positive weight constraints. We will only discuss networks with an even number of layers; however, the result is also valid for an odd number of layers.

A first crucial observation is that $\sigma$ and $\sigma'$ have the same monotonicity, but saturate in opposite directions.

**Proposition 3.8.** *If $\sigma(x)$ is monotonic non-decreasing, then its point reflection $\sigma'(x)$ is also monotonic non-decreasing. If $\sigma(x)$ saturates, then $\sigma'(x)$ also saturates but in the opposite direction.*

From Proposition 3.8 we can obtain an immediate corrollary of Theorem 3.5 which will prove useful:

**Proposition 3.9.** *An MLP with at least $4$ layers, non-negative weights, and alternating activation $\sigma$ and $\sigma'$ is a universal monotonic approximator, provided that $\sigma$ saturates on at least one side.*

The second observation is that imposing non-positive constraints in two adjacent layers with an activation function in between is equivalent to imposing non-negative constraints in the two layers and using a point-reflected activation function between them.

**Proposition 3.10.** *An MLP with $W^{(k)} \leq 0$, $W^{(k+1)} \leq 0$ and $\sigma^{(k)}(x) = \sigma(x)$, is equivalent to an MLP with $W^{(k)} \geq 0$, $W^{(k+1)} \geq 0$ and $\sigma^{(k)}(x) = \sigma'(x) = -\sigma(-x)$.*

From this, it follows that an MLP with an even number of layers, non-positive weights, and activation $\sigma$ at all layers is equivalent to an MLP with non-negative weights that alternate activations between $\sigma'$ and $\sigma$. This equivalence can be achieved using Proposition 3.10 by "flipping" the weight constraints two layers at a time, which also changes the activations at odd-numbered layers from $\sigma$ to $\sigma'$.

Thanks to Proposition 3.10, this also shows that:

**Proposition 3.11.** *If $\sigma \in \mathcal{S}^- \cup \mathcal{S}^+$, an MLP with $4$ layers, non-positive weights and activation $\sigma$, is a universal approximator for the class of monotonic functions.*

**Algorithm 1** Forward pass of a Monotonic MLP with post-activation switch

---

**Input:** data $x \in \mathbb{R}^d$, weight matrix $W \in \mathbb{R}^{d \times d'}$, bias vectors $b \in \mathbb{R}^{d'}$, monotonic activation function $\sigma$
**Output:** prediction $\hat{y} \in \mathbb{R}^{d'}$
$W^+ := \max(W, 0)$
$W^- := \min(W, 0)$
$z^+ := W^+ \sigma(x)$
$z^- := W^- \sigma(-x)$
$\hat{y} := z^+ + z^- + b$

---

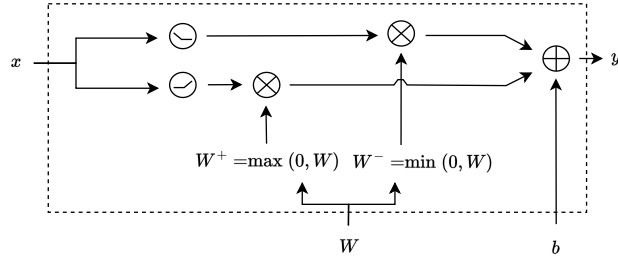

*Figure 6.* Computation graph of a single layer of a monotonic NN with the proposed learned activation via weight sign.

Similarly, we can apply the observations of this section to show that the structure proposed in Runje & Shankaranarayana (2023) can produce universal monotonic approximators using only point reflections without the need for the third activation class.

Although using Theorem 3.5 allows us to prove that a broad class of constrained MLP architectures are universal monotonic approximators, it does not necessarily translate into MLPs that are easily optimizable. Consider the computation for an arbitrary input $x$ and an MLP with ReLU activation and biases initialized to zero. If $x \geq 0$, then $-|W|x \leq 0$, and thus in the first layer $\text{ReLU}(-|W|x) = 0$. However, if $x \leq 0$, then $\text{ReLU}(-|W|x) \geq 0$ and now the second layer will saturate instead. To allow for efficient and effective optimization, we must carefully tune the initialization of the bias term to avoid having 0 gradient everywhere. Broadly speaking, the weight constraint makes the networks more sensitive to initialization.

## 4. Addressing the weight constraint

Historically, the first works that proposed a monotonic neural network formulation relied on forcing the parameters of the network to be non-negative, specifically the matrices $W$ in the affine transformations, combined with bounded monotonic activations, is a sufficient condition to guarantee that the overall function is monotonic (Daniels & Velikova, 2010; Sill & Abu-Mostafa, 1996). Recently, Runje & Shankaranarayana (2023) showed a way to build effective monotonic MLPs with such a technique by exploiting multiple activations. However, even though the use of constrained weights and bounded activation is easy to implement and can be optimized with any unconstrained gradient optimizer, poor initialization could lead to vanishing gradient dynamics, as further explored in Appendix A.2 and Appendix A.3. Instead, we will show how to address this issue while also tackling the necessity of alternating the activation saturation to have universal approximation capabilities.

### 4.1. Relaxing weight constraints with activation switches

Assuming that we used the weight-constrained formulation

for the construction proposed in Section 3.3, we would still be left to decide the sequence of activations that should be used for the MLP, which might be unclear or necessitate further hyperparameter tuning. However, by slightly rearranging the order of operations, it is possible to construct a monotone MLP that does not require manual tuning of the activation saturation side, while, at the same time, relaxing the weight constraint.

Let us thus consider a single layer $f(x) = \sigma(|W|x + b)$ in a constrained MLP, that uses the absolute value for weight reparametrization. Instead of constraining weights, we can separate $W$ into its positive and negative parts $W^+ = \max(W, 0)$ and $W^- = \min(W, 0)$. This allows us to express the affine transformation as

$$|W|x + b = W^+ x - W^- x + b. \tag{11}$$

Applying the non-linearity to each term of Equation (11) individually instead of applying it to $|W|x$, and sharing the bias term, leads to the parametrization:

$$\hat{f}(x) = \sigma(W^+ x + b) - \sigma(W^- x + b). \tag{12}$$

**Proposition 4.1.** *Any function representable using an affine transformation with non-negative weights followed by either $\sigma$ or $\sigma'$ can also be represented using Equation (12), up to a constant factor.*

*Proof.* When all entries of $W$ have the same sign, one of the two terms in Equation (12) collapses to $\pm\sigma(b)$. Specifically, when $W \geq 0$, the expression reduces to $\sigma(|W|x + b) - \sigma(b)$, while when $W \leq 0$ it reduces to $\sigma(b) - \sigma(-|W|x + b)$ instead. To conclude the proof recall that $-\sigma(-x) = \sigma'(x)$. $\square$

The additional constant factor can be accounted for in the bias term of the following layer. Therefore, Proposition 4.1 covers both cases employed in Proposition 3.9. This shows that an MLP obtained by stacking at least 4 blocks parametrized as Equation (12) is a universal approximator for monotonic functions. Hence, the proposed formulation

Table 1. Test metrics across different datasets. The best-performing architecture per dataset is bolded.

| Method | COMPAS (Test Accuracy) | Blog Feedback (Test RMSE) | Loan Defaulter (Test Accuracy) | AutoMPG (Test MSE) | Heart Disease (Test Accuracy) |
|---|---|---|---|---|---|
| XGBoost | $68.5\% \pm 0.1\%$ | $0.176 \pm 0.005$ | $63.7\% \pm 0.1\%$ | - | - |
| Certified | $68.8\% \pm 0.2\%$ | $0.159 \pm 0.001$ | $65.2\% \pm 0.1\%$ | | - |
| Non-Neg-DNN | $69.3\% \pm 0.1\%$ | $0.154 \pm 0.001$ | $65.2\% \pm 0.1\%$ | $10.31 \pm 1.86$ | $89\% \pm 1\%$ |
| DLN | $67.9\% \pm 0.3\%$ | $0.161 \pm 0.001$ | $65.1\% \pm 0.2\%$ | $13.34 \pm 2.42$ | $86\% \pm 2\%$ |
| Min-Max Net | $67.8\% \pm 0.1\%$ | $0.163 \pm 0.001$ | $64.9\% \pm 0.1\%$ | $10.14 \pm 1.54$ | $75\% \pm 4\%$ |
| Constrained MNN | $69.2\% \pm 0.2\%$ | $0.154 \pm 0.001$ | $65.3\% \pm 0.1\%$ | $8.37 \pm 0.08$ | $89\% \pm 0\%$ |
| Scalable MNN | $69.3\% \pm 0.9\%$ | $0.150 \pm 0.001$ | $65.0\% \pm 0.1\%$ | $7.44 \pm 1.20$ | $88\% \pm 4\%$ |
| Expressive MNN | $69.3\% \pm 0.1\%$ | $0.160 \pm 0.001$ | $\mathbf{65.4\% \pm 0.1\%}$ | $7.58 \pm 1.20$ | $90\% \pm 2\%$ |
| **Ours** | $\mathbf{69.5\% \pm 0.1\%}$ | $\mathbf{0.149 \pm 0.001}$ | $\mathbf{65.4\% \pm 0.1\%}$ | $\mathbf{7.34 \pm 0.46}$ | $\mathbf{94\% \pm 1\%}$ |

is more expressive compared to a simple weight constraint, given that they are only a special case of Equation (12).

Alternatively, one could apply similar reasoning working backwards from the last layer of the network, which would lead to an alternative formulation, given by

$$\hat{f}(x) = W^+\sigma(x) + W^-\sigma(-x) + b. \qquad (13)$$

We will refer to Equation (12) and to Equation (13) as pre-activation switch and post-activation switch, respectively.

In Figure 6, we report only the post-activation switch's pseudocode and computation graph since it will be the formulation that will also be employed for the experimental part of the paper. The pre-activation corresponding algorithm and computational graph can be found in Appendix A.5.

Indeed, the simplicity of the approach can be appreciated: it shares most of the steps of the forward pass of a traditional MLP due to the relaxation of the weight constraint and does not require additional special care for initializations. In Appendix A.2 we provide additional details on the optimization properties of the proposed formulation. For all experiments, the default PyTorch (Paszke et al., 2019) initialization was used without the need for additional tuning. A naive implementation can be achieved using a second matrix multiplication. This additional operation can be easily parallelized and does not require additional data transfers. For the networks tested, we did not observe any overhead in practice.

## 5. Experiments

In this section, we aim to analyze the method's performance compared to other alternatives that give monotonic guarantees. The first dataset used is COMPAS (Fabris et al., 2022). COMPAS is a dataset comprised of 13 features, 4 of which have a monotonic dependency on the classification. A

second classification dataset considered is the Heart Disease dataset. It consists of 13 features, 2 of which are monotonic with respect to the output. Lastly, we also test our method on the Loan Defaulter dataset, comprised of 28 features, 5 of which have a monotonic dependency on the prediction. To test on a regression task, we use the AutoMPG dataset, comprised of 7 features, 3 of which are monotonically decreasing with respect to the output. A second dataset for regression is the Blog Feedback dataset (Buza, 2013). Contrary to all other datasets, this dataset is composed of a very small portion of monotonic covariates. In fact, the data set consists of 280 features, of which only 8 are monotonic with respect to the output, accounting for $2.8\%$ of the total.

We compare our method with several other approaches that give monotonic guarantees by construction. In particular, we compare it to XGBoost(Chen & Guestrin, 2016) Deep Lattice Network (You et al., 2017), Min-Max Networks (Daniels & Velikova, 2010), Certified Networks (Liu et al., 2020), COMET (Sivaraman et al., 2020), Constrained Monotonic Neural Networks (Runje & Shankaranarayana, 2023), Expressive Monotonic Neural Network (Nolte et al., 2023), and Scalable Monotonic Neural Networks (Kim & Lee, 2024). With Non-Neg-DNN we refer to a naive constrained monotonic MLP using sigmoid activations. Similar results are reported in (Liu et al., 2020), though in a narrower set of datasets. In Table 1, we report the final test set metrics. For the proposed method, little to no hyperparameter tuning has been performed, as the hyperparameters found by (Runje & Shankaranarayana, 2023) worked without the need for further tuning. The proposed method matches or exceeds the performance of all other recently proposed approaches.

## 6. Conclusions and future works

In this work, we have relaxed the requirements to achieve universal approximation in monotonic MLPs with con-

strained weights. We proved that alternating saturation side in the activations is a sufficient condition to achieve this property with a finite number of layers. In addition, we show a connection between the saturation side of the activations and the sign of the weight constraint. This allows us to show that the non-positive weight constraint is, surprisingly, more expressive than the non-negative one, which can only represent convex functions. We then use this theoretical analysis to construct a novel parameterization that relaxes the weight constraint, making the network less sensitive to initialization. The activation saturation side is learnable, which ensures universal approximation capabilities even when using monotonic convex activation, which was previously not possible. MLPs built with our fully connected, monotone layer achieve state-of-the-art performance.

Although this work proves that any monotonic saturating activation can be used to build monotonic MLPs, it is still an open question whether non-saturating activations, such as Leaky-ReLU, can be used to build monotonic MLPs. Furthermore, batch normalization has proven to be highly effective in the unconstrained case. Still, it has never been used as a possible solution to the initialization problem for the monotonic case.

## 7. Ethical Considerations

The use of the COMPAS dataset in this research acknowledges its status as a common benchmark within the field of machine learning fairness studies (Angwin et al., 2022; Dressel & Farid, 2018). Recognizing the complexities and potential ethical challenges associated with such datasets, we emphasize a commitment to responsible research practices. We prioritize transparency and ethical rigor throughout our study to ensure that the methodologies employed and the conclusions drawn contribute constructively to the ongoing discourse in AI ethics and fairness. This approach underlines our dedication to advancing machine learning applications in a manner that is conscious of their broader societal impacts.

## Impact Statement

This paper presents work whose goal is to advance the field of Machine Learning. There are many potential societal consequences of our work, none of which we feel must be specifically highlighted here.

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

# A. Appendix

The appendix is structured in the following way:

- **Appendix A.1**: in this section, we show the arguably simplest though loosest bound to prove that non-negative constrained MLPs with ReLU and ReLU' activations are universal approximators.

- **Appendix A.2**: we discuss how parameterizing the network with non-negative weights leads to optimization issues, specifically vanishing gradients dynamics.

- **Appendix A.3**: in this section, we will compare the proposed method to the bounded-activation counterpart, showing how the formulation with sigmoidal activation suffers from vanishing gradients.

- **Appendix A.4**: in this section we prove the results of Theorem 3.5 for the opposite alternation case.

- **Appendix A.5**: as reported in Section 4, we propose two possible parametrizations, a pre-activation switch, and a post-activation switch. In Appendix A.5, the pseudocode and the computational graph of the two can be found.

- **Appendix A.6 and Appendix A.7**: in these sections, we report further information regarding how the results have been obtained and about the datasets employed for this work.

- **Appendix A.8**: since Theorem 3.5 only requires the non-linearity to be saturating, in this section we report a brief overview of other activations that can be applied with the proposed method, in order to underline how it is more general than just using ReLU activations.

- **Appendix A.9**: the proof provided for Theorem 3.5 is different to the ones previously proposed in literature. However, it still ends with the result of requiring 4 layers to be a universal approximator, as previously shown in (Mikulincer & Reichman, 2022) for the heavy-side function. For readers that are already familiar with such proof, we also report in Appendix A.9 a proof very similar to the one in (Mikulincer & Reichman, 2022), trying to reuse as much as possible the original structure.

## A.1. Naive bound for universal approximation of alternating MLPs

A simpler, though looser, bound to prove that MLPs with alternating ReLU and its point reflection ReLU' activations are a universal monotonic function approximator can be achieved by building on the proof of (Mikulincer & Reichman, 2022). Two simple observations are sufficient.

*Remark* A.1. the composition of ReLU and its point reflections $\text{ReLU}'(x) = -\text{ReLU}(-x)$ can approximate the threshold function $\mathbb{1}_{x \geq 0}$ arbitrarily well:

$$\lim_{\alpha \to +\infty} \text{ReLU}(\text{ReLU}'(\alpha x) + 1) = \mathbb{1}_{x \geq 0}(x) \tag{14}$$

$$\lim_{\alpha \to +\infty} \text{ReLU}'(\text{ReLU}(\alpha x) - 1) = \mathbb{1}_{x \geq 0}(x) - 1 \tag{15}$$

A representation of Equation (14) is provided in Figure 2.

The reason why we can approximate non-convex functions using only ReLU-like activations is reported in Proposition 3.2. However, considering Proposition 3.8, we can see how this limitation can be addressed.

*Remark* A.2. The formulas in Equation (14) can be implemented with a 2-layer constrained MLP, alternating ReLU and ReLU' activations.

This is enough to leverage the existing results for threshold-activated MLP (Mikulincer & Reichman, 2022). This includes the best-known bound on the number of required hidden layers, which, however, doubles from 3 to 6 due to the need for two ReLU layers for the Heavyside approximation. However, this naive bound is unnecessarily loose, as shown in Theorem 3.5.

## A.2. Initialization issues and Vanishing Gradient in Constrained MLPs

### A.2.1. VANISHING GRADIENTS DYNAMICS IN CONSTRAINED MONOTONIC MLPS WITH BOUNDED ACTIVATIONS

As reported in Section 3, a naive approach to ensure monotonicity is to have monotonic activations and to impose monotonicity to the weights, constraining them to be non-negative. For this reason, the affine transformations of these networks are usually parametrized as $l(x) = g(W)x + b$, for some transformation $g : \mathbb{R} \to \mathbb{R}_+$. Note that the bias can be any value, as it is a constant and thus does not affect the gradient.

Such networks employed bounded activations, like sigmoid, tanh, or ReLU6, to have convex-concave activations. This peculiarity makes them very sensitive to initialization and can potentially lead to vanishing gradient dynamics (Glorot & Bengio, 2010). To see why constraining weights to be non-negative exacerbates this condition, consider a monotonic MLP with sigmoidal activations, initialized with random weights according to known, widely used initializers, such as Glorot, where each matrix is sampled from a symmetric distribution around zero with some variance. Instead, the biases are initialized to zero, as is usually done. Let's assume using $g(x) = |x|$, but the same reasoning can be applied to any other mapping $g$. At this point, the MLP comprises layers of the following form $\sigma(x) = \sigma(|W|x + b)$. Now, let's consider the second layer of such MLP. Since the first layer has applied the sigmoid activation, then $\sigma^{(1)}(x) \in (0, 1)$. Because of this, $|W^{(2)}|\sigma^{(1)}(x)$ will be a product of all non-negative terms. Therefore, its result can become significantly large. Then, when applying the sigmoid activation of the second layer, it will most likely saturate due to the large positive values returned from the affine transformation. Going on with this reasoning for multiple layers, such behavior will be exacerbated. Appendix A.3 shows one example of such behavior for a very simple function. The same behavior occurs for ReLU6 MLPs, where the gradient might even become exactly 0, and for tanh MLPs if, for example, the dataset is normalized, which is one of the most commonly used data-preprocessing.

A simple evidence of such dynamic can be seen in Appendix A.3, where even with a 1D toy example, constrained monotonic MLPs with sigmoid with few layers show signs of vanishing gradient (Figure 8). However, such signs of vanishing gradient are evident since initialization, as shown in Figure 9

One possible solution might be using BatchNormalization layers (Ioffe, 2015). BatchNorm has already shown its effectiveness in tackling initialization and optimization problems. Indeed, BatchNorm is comprised only of the following transformation:

$$BN(x) = \frac{x - \mathbb{E}[x]}{\sqrt{\text{Var}[x] + \epsilon}} \cdot \gamma + \beta$$

Considering that $\sqrt{\text{Var}[x] + \epsilon} > 0$, forcing $\gamma \geq 0$ by construction, for example, using $\gamma = \text{SoftPlus}(\gamma')$, makes such operation monotonic. Usually, it is initialized as $\beta = 0$ and $\gamma = 1$. For this reason, if used as a pre-activation layer, it might

address exploding pre-activation values, standardizing them around zero. However, the investigation of this approach falls out of the scope of this work, and it's left as a future line of research.

### A.2.2. INITIALIZATION OF SWITCH MONOTONIC NEURAL NETWORKS

The proposed activation switch parametrization shown in Equation (12) and Equation (13) aims at relaxing monotonic MLPs by the weight constraint required to ensure monotonicity, while also removing the need to pick a correct sequence of activations manually. As addressed in the previous chapter, that weight constraint easily leads to vanishing gradients if used in conjunction with bounded activation, extremized when using ReLU6 activations, which leads almost always to a completely dead MLP, as reported in Appendix A.3. Therefore, our parametrization tries to solve this problem from two points: avoiding directly constraining weights and using unbounded activations.

Nonetheless, it still deviates from the well-studied original MLP formulation. In particular, separating the $W$ matrix into positive and negative parts might still introduce initialization issues.

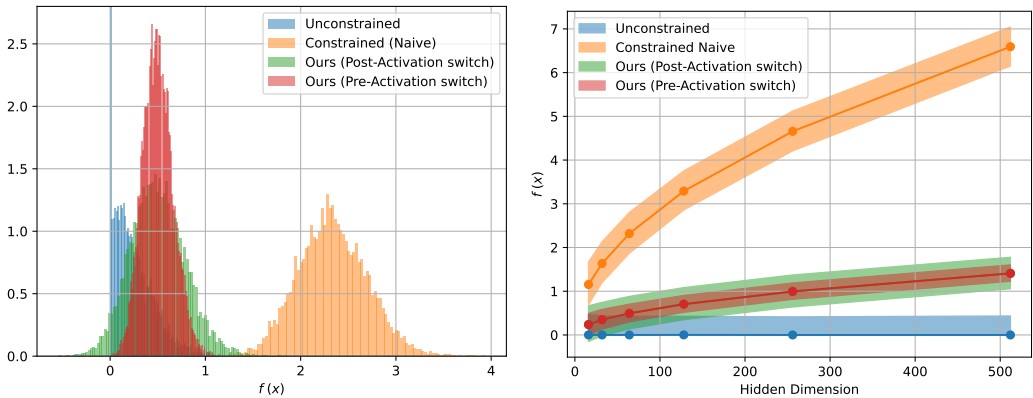

*Figure 7.* First plot, the distribution of the output of an MLP with the different parametrizations. Second plot, the scaling law of the expected output after initialization of the different parametrization. In both images, it can be seen how the naive constrained MLP has a very different scaling behavior compared to the rest.

From an empirical perspective, in Figure 7 we can observe the distribution of a the output of a multilayer MLP with different parametrization. Unconstrained refers to a naive MLP, Constrained Naive refers to an MLP with $|W|$ as weight parameterization, while pre/post activation switch refers to Equation (12) and Equation (13). Indeed, it can be seen how the naive MLP, no matter the hidden dimension, have 0-mean in predicted value. Instead, the switch activation has a very slow tendency to increase the expected output from random initialization, as predicted by theory. However, such slight increase, is notably largely smaller than the one induced by naively constraining the weights to be positive.

However, empirically, the activation-switch formulation does not exhibit initialization issues. Indeed, the result reported in Table 1 have been obtained with the default PyTorch initialization. Furthermore, such results are the aggregation of multiple seeds. Thus, the networks used have been initialized with different values.

Overall, we can see how, even though the behavior of activation-switch parametrization slightly deviates from the one by a normal MLP, such difference does not hinder many performance while still being less than the one obtained by the constrained counterpart that does not use such a trick.

### A.3. Comparison of different parametrizations on trivial datasets

To showcase the effectiveness of the proposed method to the bounded-activation counterpart, in Figure 8 we compare them on a simple synthetic example. In particular, the models are asked to approximate $f(x) = \cos(x) + x$, a simple 1D monotonic function with multiple saddle points. For this reason, it is fundamental for the approximation model to be very flexible. To showcase the different performances, we will test 4 models. The first model to test is an unconstrained NN, which shows that an unconstrained model can learn such a function. The second model is a monotonic NN with non-negative and ReLU activations, which shows that, as shown in theory, it cannot approximate a nonconvex function. The third model is a monotonic NN with non-negative and sigmoid activations. This model, instead, is shown to be a universal approximator for monotonic functions but suffers from vanishing gradients. Lastly, the fourth model is the proposed parametrization, specifically the post-activation setting, as described in Section 4.1.

In Figure 8, it can be seen how the model with non-negative and ReLU activations cannot learn the function as predicted by theory, since the function that is asked to learn is non-convex. Instead, both the sigmoid model and our proposed approach successfully approximate it. Still, the sigmoid function struggles to be optimized due to the complications of using sigmoid activations. Instead, the proposed method exploits rectified linear activations, which, under a regime where the number of dead neurons is not too high, is much easier to optimize, as explained in the original work that introduced such activation Glorot & Bengio (2010) and Raghu et al. (2017).

Such a difference is also evident in analyzing the Negative Log Likelihood (NLL) loss of the training. We report in Figure 8 the various training losses obtained with two different sizes of layers. The naive monotonic ReLU, which cannot approximate such a function, is indeed the worst. However, even though the sigmoid monotonic NN is a universal approximator, it is the slowest to learn, probably due to the vanishing gradient problem. Instead, the proposed method that uses ReLU activations is the fastest to converge, almost catching the unconstrained model in the setting with more neurons.

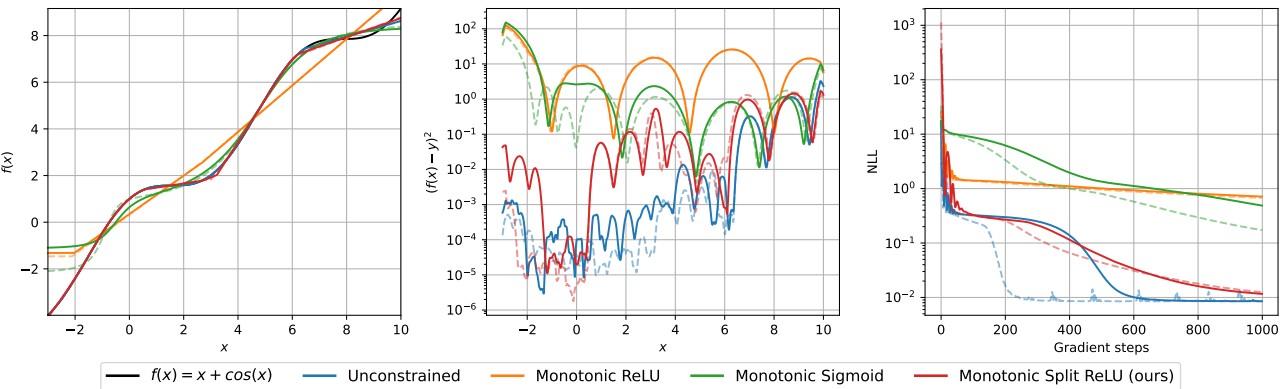

*Figure 8.* First plot, approximation of $f(x)$ using MLPs with layers of 128 neurons. Second plot, approximation of $f(x)$ using MLPs with layers of 256 neurons. Last plot, training losses of the different methods (full lines represent versions with 128 neurons, dashed lines represent versions with 256 neurons).

Generally speaking, as also reported at the end of Section 3.3, MLPs with constrained weights require a careful initialization to avoid non-optimizable configurations. The proposed method in Appendix A.5 alleviates this behavior but is not indifferent to it.

In order to showcase the vanishing gradient problem exacerbated by the non-negatively constraining, in Figure 9 we create a 128-neuron wide MLP with varying numbers of hidden layers, and we compare the average gradient of the output with respect to the parameters on the same function approximation problem presented earlier in Figure 8. It can be observed how the sigmoid monotonic MLP, even with a small number of layers, has one order of magnitude less gradient magnitude; in particular, it has an average gradient of $0.0019$ for 4 layers and $0.00099$ for 10 layers. Instead, the ReLU monotonic MLP has an exploding gradient due to the accumulation of activations induced by the pairing of ReLU activation and positive weight; in particular, it starts from a gradient magnitude of $3.54$ for 4 layers and goes to $3311.00$ for 10 layers. Finally, the proposed approach keeps the gradient magnitude in a reasonable magnitude range, starting from a gradient of $0.010$ for 4 layers and going to $1.259$ for 10 layers. Results are averaged over 20 different random initializations, and plot shows $\pm 1\sigma$.

In order to better analyze the optimization problems of these architectures, we also report in Figure 10 the distributions of the gradients of a 6-layers MLP with the various architectures. It can be seen that the sigmoid MLP has extremely low gradients for the initial layers, leading to slow learning. On the other hand, the ReLU MLP has exploding gradients for the final layers.

It is worth noting that the same exact configuration, using ReLU6 as activation, leads to a dead network, as all gradients are zeroed out due to the saturated section of the activation.

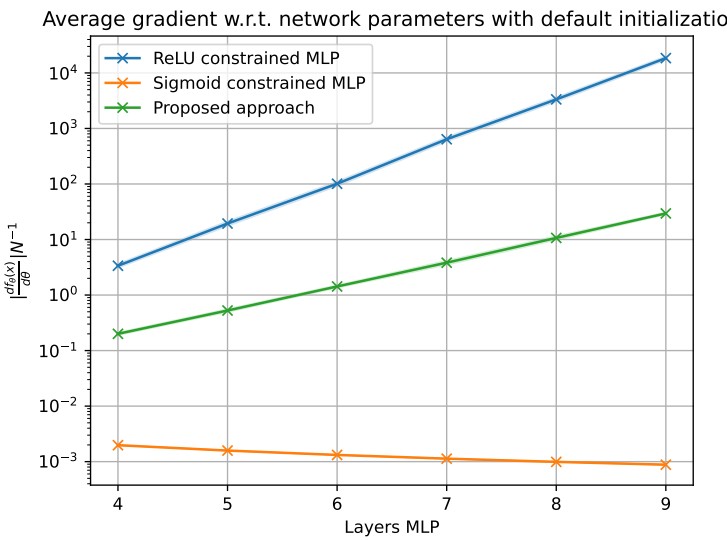

*Figure 9.* Average gradient from monotonic MLPs varying the number of layers. Data is shown in the log scale for the $y$-axis.

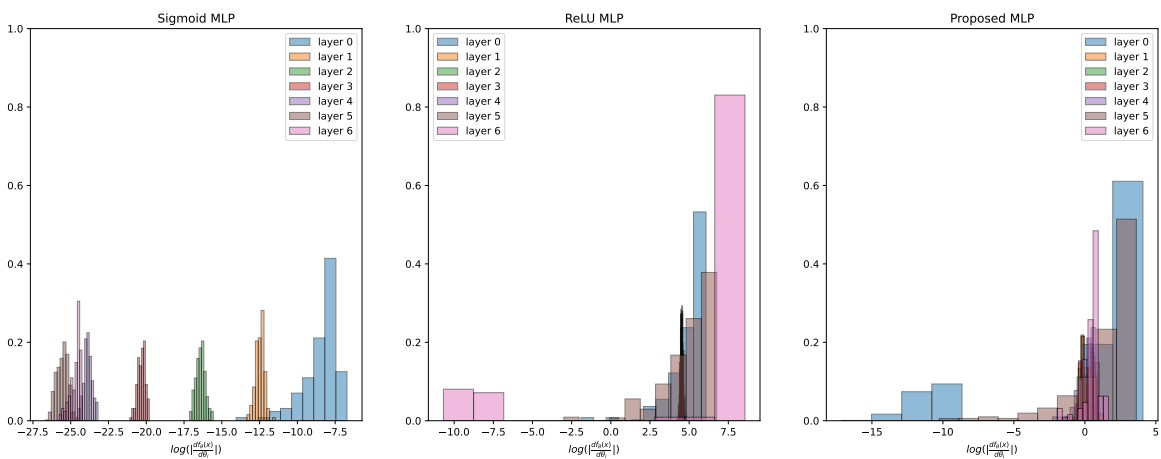

*Figure 10.* Distribution of gradients from monotonic MLPs for each layer (layer 0 is the final one, layer 6 is the first after the input).

## A.4. Proof for opposite alternation of activation for Theorem 3.5

In this section, we will conclude the proof of Theorem 3.5, considering the case with activations that alternate in the opposite direction from the one reported in the main text. Indeed, in Section 3.2 we proved the result for the case with $\sigma^{(1)} \in \mathcal{S}^-, \sigma^{(2)} \in \mathcal{S}^+, \sigma^{(3)} \in \mathcal{S}^-$, while in this section we will prove the case with $\sigma^{(1)} \in \mathcal{S}^+, \sigma^{(2)} \in \mathcal{S}^-, \sigma^{(3)} \in \mathcal{S}^+$. The proof is extremely similar, with just a few opposite signs due to the opposite alternation. Thus, most constructions will be shared.

*Proof of Theorem 3.5 with opposite alternation.* Assume, without loss of generality, that the points $x_1, \ldots, x_n$ are ordered so that $i < j \implies f(x_i) \leq f(x_j)$, with ties resolved arbitrarly. We will proceed by construction, layer by layer.

**Layer 1** Since the function to interpolate is monotonic, for any couple of points $i < j : f(x_i) < f(x_j)$ it is possible to find a hyperplane with non-negative normal, with positive and negative half spaces denoted by $A_{j/i}^+$ and $A_{j/i}^-$, such that $x_i \in A_{j/i}^-, x_j \in A_{j/i}^+$.

Using Lemma 3.6, we can ensure that it is possible to have:

$$\begin{cases} h_i^{(1)}(x) \approx \sigma^{(1)}(+\infty) = 0, & \text{if } x \in A_{j/i}^+ \\ h_i^{(1)}(x) \approx \sigma^{(1)}(-\infty) < 0, & \text{otherwise} \end{cases} \tag{16}$$

**Layer 2** Let us construct the set $A_i^{(2)} = \bigcap_{j:j<i} A_{i/j}^+$. Note that the sets $A_i^{(2)}$ always contain $x_i$ and do not contain any $x_j$ for $j < i$. Using Equation (16), we can apply Lemma 3.7, which ensures that it is possible to have the following[5]:

$$\begin{cases} h_i^{(2)}(x) \approx 0, & \text{if } x \in A_i^{(2)} \\ h_i^{(2)}(x) \approx \gamma^{(2)} > 0, & \text{otherwise} \end{cases} \tag{17}$$

**Layer 3** Consider $A_i^{(3)} = \bigcap_{j:j>i} \bar{A}_j^{(2)}$, where $\bar{A}_j^{(2)}$ is the complement of $A_j^{(2)}$. Using Equation (17) we can once again apply Lemma 3.7, which ensures that it is possible to have the following[6]:

$$h_i^{(3)}(x) \approx \gamma^{(3)} \mathbb{1}_{A_i^{(3)}}(x) \tag{18}$$

Now, we will show that $A_i^{(3)}$ represents a level set, i.e. $x_j \in A_i^{(3)} \iff f(x_j) \leq f(x_i)$. To do so, consider that $\bar{A}_i^{(3)} = \bigcup_{j:j>i} A_j^{(2)}$. Since $x_j \in A_j^{(2)}$, then $x_j \in \bar{A}_i^{(3)}$ for $j > i$. Similarly since $x_j$ is the smallest point contained in $A_j^{(2)}$, $\bar{A}_i^{(3)}$ cannot contain $x_i$ or any point smaller than $x_i$. This shows that $A_i^{(3)}$ contains exactly the points $\{x_j : f(x_j) \leq f(x_i)\}$.

**Layer 4** To conclude the proof, simply take the weights at the fourth layer to be :

$$w = \left[ \frac{f(x_1) - f(x_2)}{\gamma^{(3)}}, \ldots, \frac{f(x_{n-1}) - f(x_n)}{\gamma^{(3)}}, \frac{f(x_n) - b}{\gamma^{(3)}} \right]$$

Note that compared to Equation (8), here $\gamma^{(3)}$ is now negative, and the terms in the numerators' difference are reversed. Since the points are ordered, this ensures that $w$ contains all non-negative terms, when bias term $b$ is taken to be $b \geq f(x_n)$. Defining $f(x_{n+1}) = b$, the output of the MLP can be expressed as:

$$g_\theta(x) = w^T h^{(3)}(x) + b = b + \sum_{j=1}^n (f(x_j) - f(x_{j+1})) \mathbb{1}_{A_j^{(3)}}(x) \tag{19}$$

Evaluating Equation (19) at any of the points $x_i$, it reduces to the telescopic sum:

$$g_\theta(x_i) = f(x_n) + \sum_{j=i}^{n-1} (f(x_j) - f(x_{j+1})) = f(x_i) \tag{20}$$

Thus proving that the network correctly interpolates the target function. $\square$

---

[5]In this case $\gamma^{(2)} > 0$ since we are considering the case where $\sigma^{(2)}$ saturates left.
[6]In this case $\gamma^{(3)} < 0$ since we are considering the case where $\sigma^{(3)}$ saturates right.

## A.5. Algorithms

In Section 4, we show how we can parametrize the activation switch using the sign of the weights. For such a mechanism, we propose two different parametrizations, one where the switch is applied post-activation and another one pre-activation. In Figure 6, we report both the pseudo-code and the computational graph for the post-activation formulation. For completeness, in this section, we also report the pre-activation pseudo-code and computational graph, and for readability and to ease the comparison, we report them side by side, reporting again the post-activation formulation also reported in the main text. In particular, in Figure 11, we report the two pseudocode side-by-side, and in Appendix A.5, the relative pseudocodes.

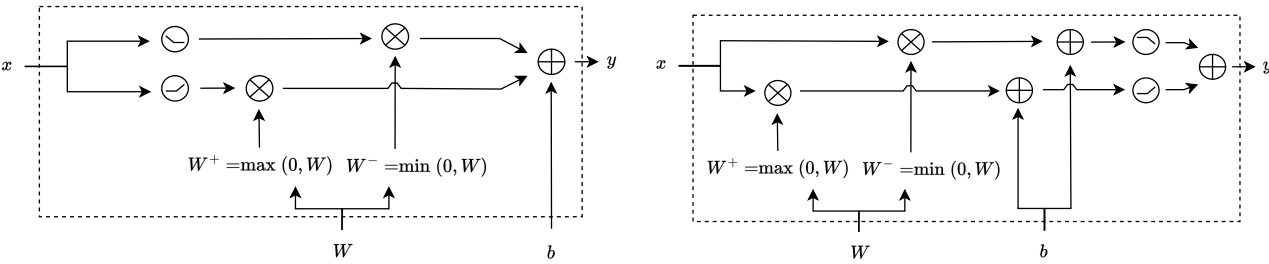

*Figure 11.* Computation graph of a single layer of a ReLU monotonic NN with the proposed learned activation via weight sign. The left plot reports the computational graph of the post-activation, and the right plot shows the pre-activation switch.

---

**Algorithm 2** Forward pass of a Monotonic ReLU MLP with pre-activation switch

---

**Input:** data $x \in \mathbb{R}^n$, weight matrix $W \in \mathbb{R}^{h_l \times h_{l-1}}$, bias vectors $b \in \mathbb{R}^{h_l}$, activation function $\sigma$
**Output:** prediction $\hat{y} \in \mathbb{R}^{h_L}$
$W^+ := \max(W, 0)$
$W^- := \min(W, 0)$
$z^+ := W^+ x + b$
$z^- := W^- x + b$
$\hat{y} := \sigma(z^+) - \sigma(z^-)$

---

**Algorithm 3** Forward pass of a Monotonic ReLU MLP with post-activation switch

---

**Input:** data $x \in \mathbb{R}^n$, weight matrix $W \in \mathbb{R}^{h_l \times h_{l-1}}$, bias vectors $b \in \mathbb{R}^{h_l}$, activation function $\sigma$
**Output:** prediction $\hat{y} \in \mathbb{R}^{h_L}$
$W^+ := \max(W, 0)$
$W^- := \min(W, 0)$
$z^+ := W^+ \sigma(x)$
$z^- := W^- \sigma(-x)$
$\hat{y} := z^+ + z^- + b$

---

## A.6. Dataset description

For this work, the code was heavily based on the code provided by Runje & Shankaranarayana (2023) in order to ensure that the used dataset matched exactly. For this reason, we will report a short description of the employed dataset, but for a further and more detailed description, refer to the original work (Runje & Shankaranarayana, 2023).

- **COMPAS**: This dataset is a binary classification dataset composed of criminal records, comprised of 13 features, 4 of which are monotonic.

- **Blog Feedback**: This dataset is a regression dataset comprised of 280 features, 8 of which are monotonic, aimed at predicting the number of comments within 24h.

- **Auto MPG**: This dataset is a regression dataset aimed at predicting the miles-per-gallon consumption and is comprised of 7 features, 3 of which are monotonic.

- **Heart Disease**: This dataset is a classification dataset composed of 13 features, 2 of which are monotonic, aimed at predicting a possible heart disease.

- **Loan Defaulter**: This dataset is a classification dataset composed of 28 features, 5 of which are monotonic, and is aimed at predicting loan defaults.

### A.7. Experiments description

The following are the specifications used to obtain the results reported in Table 2. The experiments were developed in PyTorch. The training was performed using the Adam optimizer implementation from the PyTorch Library. For all datasets except BlogFeedback, no hyperparameter tuning has been carried out. Instead, we used similar networks in size and architecture to (Runje & Shankaranarayana, 2023). For BlogFeedback instead, special care was required, as the dataset is very large, but the features are very sparse and mostly unconstrained (only 3% are monotonic). Therefore, a small hyperparameter tuning has been done to find the best setting. CELU has been used as an activation function for the smaller MLPs to avoid dead neurons.

*Table 2.* Hyper-parameters used for results reported in Table 1

| Hyper-parameter | COMPAS | Blog Feedback | Loan Defaulter | AutoMPG | Heart Disease |
|---|---|---|---|---|---|
| Learning-rate | $10^{-3}$ | $10^{-2}$ | $10^{-3}$ | $10^{-3}$ | $10^{-3}$ |
| Epochs | 100 | 1000 | 50 | 300 | 300 |
| Batch-size | 8 | 256 | 256 | 8 | 8 |
| Free layers size | 16 | 2 | 16 | 8 | 16 |
| Number of free layers | 3 | 2 | 3 | 3 | 3 |
| Monotonic layers size | 16 | 3 | 16 | 8 | 16 |
| Number of monotonic layers | 3 | 2 | 3 | 3 | 3 |
| Activation | ReLU | CELU | ReLU | CELU | ReLU |

## A.8. Extension to other activations

In the rest of the paper, for all the practical examples, we assumed that ReLU was the activation chosen for the MLP. However, the results in Section 3.3 and Section 4.1 only require that the activation function saturates in at least one of the two sides, other than being monotonic. If ReLU falls in such a category, it is not the only one, and many other widely used ReLU-like activations satisfy the minimal assumptions of Theorem 3.5. For this reason, we will now analyze many other activations and report whether they comply with our construction. In particular, we report in Table 3 multiple widely used activations. With them, we also report the respective gradients, whether they are non-decreasing and saturating, and whether they can be used for the proposed approach.

It can be seen that the proposed method allows the usage of most of today's widely used activations. However, it is crucial to notice that even though the proposed method allows for saturating activations, it also can be used with bounded activations, such as sigmoid and tanh, but that might bring almost no additional advantage over the weight-constrained counterpart. Any activation that saturates at least one side can be used, given that it is monotonic. Still, the real advantage comes from activations that saturate only one side.

*Table 3.* Widely used activations with their corresponding properties, and whether they can be used or not.

| Name | Function | Gradient | Monotone | Saturates | Usable |
|---|---|---|---|---|---|
| ReLU | $\begin{cases} x & \text{if } x \geq 0 \\ 0 & \text{otherwise} \end{cases}$ | $\begin{cases} 1 & \text{if } x \geq 0 \\ 0 & \text{otherwise} \end{cases}$ | ✓ | ✓ | ✓ |
| LeakyReLU | $\begin{cases} x & \text{if } x \geq 0 \\ \alpha x & \text{otherwise} \end{cases}$ | $\begin{cases} 1 & \text{if } x \geq 0 \\ \alpha & \text{otherwise} \end{cases}$ | ✓[1] | ✗ | ✗ |
| PReLU | $\begin{cases} x & \text{if } x \geq 0 \\ \alpha x & \text{otherwise} \end{cases}$ ($\alpha$ learnable) | $\begin{cases} 1 & \text{if } x \geq 0 \\ \alpha & \text{otherwise} \end{cases}$ | ✓[1] | ✓ | ✓[1] |
| ReLU6 | $\begin{cases} 6 & \text{if } x \geq 6 \\ x & \text{if } 0 \leq x \leq 6 \\ 0 & \text{otherwise} \end{cases}$ | $\begin{cases} 0 & \text{if } x \geq 6 \\ 1 & \text{if } 0 \leq x \leq 6 \\ 0 & \text{otherwise} \end{cases}$ | ✓ | ✓ | ✓ |
| ELU | $\begin{cases} x & \text{if } x \geq 0 \\ \alpha(e^x - 1) & \text{otherwise} \end{cases}$ | $\begin{cases} 1 & \text{if } x \geq 0 \\ \alpha e^x & \text{otherwise} \end{cases}$ | ✓[1] | ✓ | ✓[1] |
| SELU | $\lambda \begin{cases} x & \text{if } x \geq 0 \\ \alpha(e^x - 1) & \text{otherwise} \end{cases}$ | $\lambda \begin{cases} 1 & \text{if } x \geq 0 \\ \alpha e^x & \text{otherwise} \end{cases}$ | ✓[1] | ✓ | ✓[1] |
| CELU | $\lambda \begin{cases} x & \text{if } x \geq 0 \\ \alpha(\frac{e^x}{\alpha} - 1) & \text{otherwise} \end{cases}$ | $\lambda \begin{cases} 1 & \text{if } x \geq 0 \\ \frac{e^x}{\alpha} & \text{otherwise} \end{cases}$ | ✓[1] | ✓ | ✓[1] |
| GeLU | $x\Phi(x)$ | $\Phi(x)\frac{1}{\sqrt{2\pi}}e^{\frac{-x^2}{2}}$ | ✗ | ✓ | ✗ |
| SiLU/Swish | $x\sigma(x)$ | $\frac{e^x(x+e^x+1)}{(e^x+1)^2}$ | ✗ | ✓ | ✗ |
| Sigmoid | $\frac{1}{1+e^{-x}}$ | $\frac{e^{-x}}{(1+e^{-x})^2}$ | ✓ | ✓ | ✓ |
| Tanh | $\frac{e^x-e^{-x}}{e^x+e^{-x}}$ | $1 - \left(\frac{e^x-e^{-x}}{e^x+e^{-x}}\right)^2$ | ✓ | ✓ | ✓ |
| Exp | $e^x$ | $e^x$ | ✓ | ✓ | ✓ |
| SoftSign | $\frac{x}{|x|+1}$ | $\frac{1}{(|x|+1)^2}$ | ✓ | ✓ | ✓ |
| Softplus | $\log(1 + e^x)$ | $\frac{e^x}{e^x+1}$ | ✓ | ✓ | ✓ |
| LogSigmoid | $-\log(1 + e^{-x})$ | $\frac{1}{1+e^x}$ | ✓ | ✓ | ✓ |

[1]: true only if parametrized in such a way to guarantee $\alpha \geq 0$

## A.9. Alternative proof of Theorem 3.5

In this section, we will construct a proof similar to the one proposed by Mikulincer & Reichman (2022) to prove the constant bound of required layers for a constrained MLP with Heavyside activations.

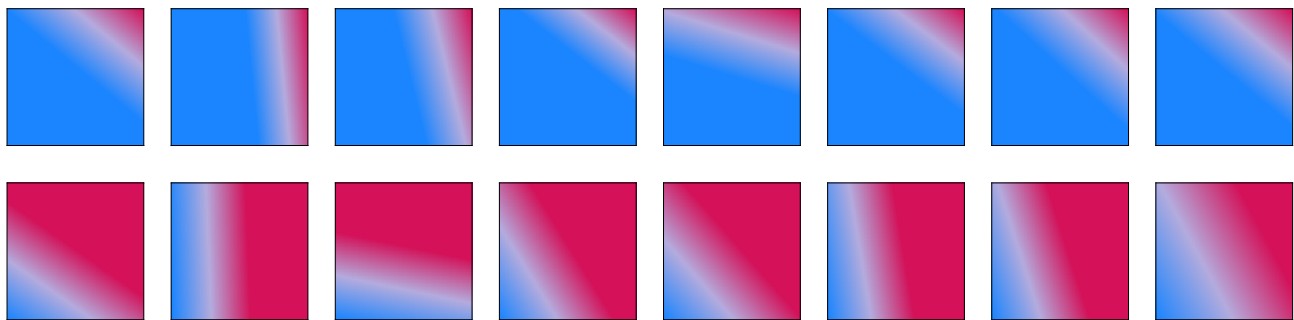

*Figure 12.* Examples of learnable functions at the first hidden layer.

**First layer construction**  First, let us show that the network can represent piece-wise functions at the first hidden layer.

**Lemma A.3.** *Consider an hyperplane defined by $\alpha^T (x - \beta) = 0$, $\alpha \in \mathbb{R}_+^k$ and $\beta \in \mathbb{R}^k$, and the open half-spaces:*

$$A^+ = \{x : \alpha^T (x - \beta) > 0\}, \tag{21}$$
$$A^- = \{x : \alpha^T (x - \beta) < 0\}. \tag{22}$$

*A single neuron in the first hidden layer of an MLP with non-negative weights can approximate [7]:*

$$h^{(1)}(x) \approx \begin{cases} \sigma^{(1)}(+\infty), & \text{if } x \in A^+ \\ \sigma^{(1)}(-\infty), & \text{if } x \in A^- \\ \sigma^{(1)}(0), & \text{otherwise} \end{cases}$$

*Proof.* Denote by $w$ the weights and by $b$ the bias associated with the hidden unit in consideration. Then, for any $\lambda \in \mathbb{R}_+$, setting the parameters to $w = \lambda \alpha^T$ and $b = \lambda \alpha^T \beta$ we have that:

$$h = \sigma^{(1)} (wx + b) = \sigma^{(1)} \left(\lambda \alpha^T (x - \beta)\right)$$

in the limit, we get:

$$h^{(1)}(x) \approx \lim_{\lambda \to +\infty} \sigma^{(1)} \left(\lambda \alpha^T (x - \beta)\right)$$

The limit is either $\sigma^{(1)}(+\infty)$, $\sigma^{(1)}(-\infty)$ or $\sigma^{(1)}(0)$ depending on the sign of $\alpha^T (x - \beta)$, proving that

$$h^{(1)}(x) \approx \begin{cases} \sigma^{(1)}(+\infty), & \text{if } \alpha^T (x - \beta) > 0 \\ \sigma^{(1)}(-\infty), & \text{if } \alpha^T (x - \beta) < 0 \\ \sigma^{(1)}(0), & \text{if } \alpha^T (x - \beta) = 0 \end{cases}$$

$\square$

For an easier interpretation of the just stated construction, we show in Figure 12 some samples from the family of functions that can be learned with this first hidden layer.

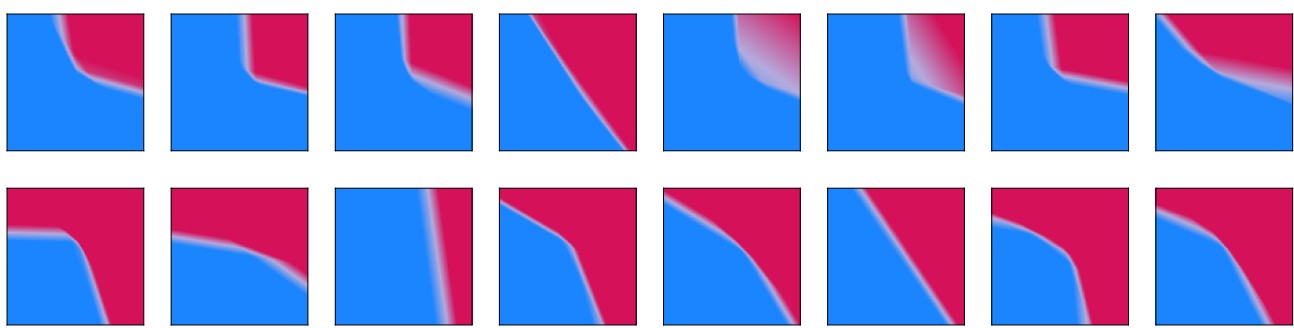

*Figure 13.* Examples of learnable indicator functions at the second hidden layer.

**Second layer construction**   Using Lemma A.3, we can show that alternating saturation directions in the activations is sufficient to represent indicator functions of intersections and unions of positive half-spaces.

**Lemma A.4.** *If $\sigma^{(1)} \in \mathcal{S}^+, \sigma^{(2)} \in \mathcal{S}^-$, there exists a rescaling factor $\gamma \in \mathbb{R}_+$ such that a single unit in the second hidden layer of an MLP with non-negative weights, can approximate:*

$$h^{(2)}(x) \approx +\gamma \mathbb{1}_{A^\cap}(x)$$

*for any $A^\cap = \bigcap_{i=1}^n A_i^+$.*

*Similarly, if $\sigma^{(1)} \in \mathcal{S}^-, \sigma^{(2)} \in \mathcal{S}^+$, it can approximate*

$$h^{(2)}(x) \approx +\gamma \mathbb{1}_{A^\cup}(x) - \gamma$$

*for any $A^\cup = \bigcup_{i=1}^n A_i^+$.*

*Proof.*   Denote by $w$ the weights and by $b$ the bias associated to the hidden unit in consideration at the second layer. For any $\lambda \in \mathbb{R}_+$, setting the weights to $w = \lambda \mathbf{1}^T$ we have that

$$h^{(2)}(x) = \sigma^{(2)}\left(wh^{(1)} + b\right) = \sigma^{(2)}\left(b + \lambda \sum_i h_i^{(1)}\right)$$

Taking the limit, the result only depends on the sign of $\sum_i h_i^{(1)}$. Using Lemma A.3, we can ensure that it is possible to have

$$h_i^{(1)}(x) \approx \begin{cases} \sigma^{(1)}(+\infty), & \text{if } x \in A_i^+ \\ \sigma^{(1)}(-\infty), & \text{if } x \in A_i^- \end{cases}$$

From here, there are two cases, depending on the saturation of the activations. We will only prove the case when the activations saturate to zero to avoid needlessly complicated formulas. However, the result holds even in the general case.

If we assume $\sigma^{(1)} \in \mathcal{S}^+, \sigma^{(2)} \in \mathcal{S}^-$:
For $x \in \bigcap_{i=1}^n A_i^+$, we have $h_i^{(1)}(x) = \sigma^{(1)}(+\infty) = 0$, while for $x \notin \bigcap_{i=1}^n A_i^+$ have $h_i^{(1)}(x) < \sigma^{(1)}(+\infty) = 0$. Therefore

$$\lim_{\lambda \to +\infty} h^{(2)}(x) \begin{cases} \sigma^{(2)}(b) = \gamma, & \text{if } x \in \bigcap_{i=1}^n A_i^+, \\ \sigma^{(2)}(-\infty) = 0, & \text{otherwise} \end{cases}$$

where $\gamma$ can be any element of the image of $\sigma^{(2)}$, which is a non negative function. Therefore for $A^\cap = \bigcap_{i=1}^n A_i^+$

$$h^{(2)}(x) \approx \gamma \mathbb{1}_{A^\cap}(x).$$

---

[7]Note that $\sigma^{(1)}(\pm\infty)$ needs not be finite.

If instead we assume $\sigma^{(1)} \in \mathcal{S}^-, \sigma^{(2)} \in \mathcal{S}^+$:

For $x \in \bigcap_{i=1}^n A_i^-$, we have $h_i^{(1)}(x) = \sigma^{(1)}(-\infty) = 0$, while for $x \in \bigcup_{i=1}^n A_i^-$ have $h_i^{(1)}(x) > 0$.

$$\lim_{\lambda \to +\infty} h^{(2)}(x) \begin{cases} \sigma^{(2)}(b) = -\gamma, & \text{if } x \notin \bigcup_{i=1}^n A_i^+, \\ \sigma^{(2)}(+\infty) = 0, & \text{otherwise} \end{cases}$$

where $-\gamma$ can be any element of the image of $\sigma^{(2)}$, that is now a non positive function. Therefore for $A^\cup = \bigcup_{i=1}^n A_i^+$

$$h^{(2)}(x) \approx -\gamma(1 - \mathbb{1}_{A^\cup}(x)) = \gamma\mathbb{1}_{A^\cup}(x) - \gamma$$

$\square$

For a more intuitive understanding of the class of functions that such a constructed second layer can learn, in Figure 13, we report some samples from that class of functions.

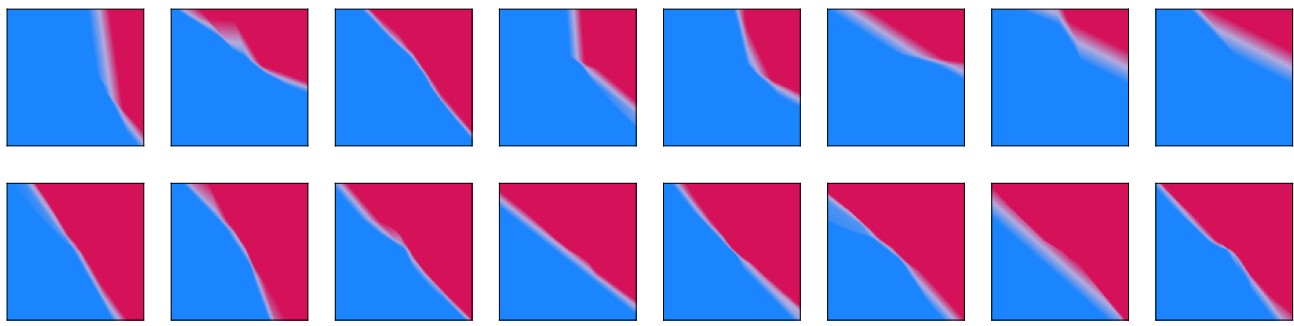

*Figure 14.* Examples of learnable functions at the third hidden layer.

**Third layer construction** Finally, let us show that a hidden unit in the third layer can perform union and intersection operations when the second-layer representations are indicator functions of sets.

**Lemma A.5.** *If $h_i^{(2)}(x) = \gamma\mathbb{1}_{A_i}$, there exists a rescaling factor $\delta \in \mathbb{R}_+$ such that a single unit in the third hidden layer of an MLP with non-negative weights can approximate:*

$$h^{(3)}(x) \approx +\delta\mathbb{1}_A(x)$$

*for any $A = \bigcup_{i=1}^n A_i$ when $\sigma^{(3)} \in \mathcal{S}^+$, and for any $A = \bigcup_{i=1}^n A_i$ if $\sigma^{(3)} \in \mathcal{S}^-$*

We are finally ready to prove the main result.

*Proof of Theorem 3.5.* Since the function to interpolate is monotonic, for any couple of points $x_i < x_j : f(x_i) < f(x_j)$ it is possible to find a hyperplane with non-negative normal, with positive and negative half spaces denoted by $A_{j/i}^+$ and $A_{j/i}^-$, such that $x_i \in A_{j/i}^-, x_j \in A_{j/i}^+$.

Let us now construct the sets:

$$A_{x_i}^\cap = \bigcap_{j:x_j<x_i} A_{i/j}^+ \tag{23}$$

$$A_{x_i}^\cup = \bigcup_{j:x_j>x_i} A_{i/j}^+ \tag{24}$$

This ensures that $x_j < x_i \implies x_j \notin A_{x_i}^\cap$. Also, since $A_{x_i}^\cap$ is obtained from the intersection of positive half-spaces, Lemma A.4 ensures a hidden unit at the second hidden layer is able to learn $h^{(2)}(x) \approx \mathbb{1}_{A_{x_i}^\cap}(x)$. Now note that $A_{x_i} = \bigcup_{j:f(x_j)>f(x_i)} A_{x_j}^\cap$ contains only and all points $x_j$ such that $f(x_j) \geq f(x_i)$. Moreover, from Lemma A.5, we know that hidden units in the third layer can approximate $\mathbb{1}_{A_{x_i}}$.

As per the previous layers, we show in Figure 14 some samples of functions that the third layer, constructed as just reported, can learn.

**Fourth layer construction** To conclude the proof, take the last layer parameters to be $w_i^{(4)} = f(x_{i+1}) - f(x_i), b^{(4)} = f(x_1)$. This produces the following function approximation

$$\bar{f}(x) = f(x_1) + \sum_i \mathbb{1}_{A_{x_i}}(f(x_{i+1}) - f(x_i))$$

. $\bar{f}(x)$ evaluated at any of the points $x_i$ provides a telescopic sum where all the terms elide, leaving $\bar{f}(x_i) = f(x_i)$. For the opposite activation pattern, the same result can be obtained in a similar fashion, considering intersections of $A_{x_i}^{\cup} = \bigcup_{j:x_j > x_i} A_{i/j}^+$ instead. $\qquad\square$

