# OpenReview forum: "Advancing Constrained Monotonic Neural Networks: Achieving Universal Approximation Beyond Bounded Activations"
_ICML.cc/2025/Conference — ICML 2025 poster_

### Official Review · Reviewer_C9i1 · 2025-02-28

**Overall Recommendation:** 4

**Summary:**

Authores generalize previous theoretical results, showing that MLPs with non-negative weight constraint and activations that saturate on alternating sides are universal approximators for monotonic functions. Additionally, they show an equivalence between saturation side in the activations and sign of the weight constraint. This allows them to prove that MLPs with convex monotone activations and non-positive constrained weights also qualify as universal approximators, in contrast to their non-negative constrained counterparts.

Experimental evaluation reinforce the validity of the theoretical results, showing that their approach compares favorably to traditional monotonic architectures

**Claims And Evidence:**

Claims made in the submission are supported by clear and convincing evidence.

**Essential References Not Discussed:**

I am not familiar enough with the broader Scientific Literature in thsi area to give a statement here.

**Experimental Designs Or Analyses:**

Yes

**Methods And Evaluation Criteria:**

Proposed methods and/or evaluation criteria make sense for the problem or application at hand.

**Other Comments Or Suggestions:**

For $h_j$ and $ A_{j/  i}$  it would be nice if $j$ would ne defined. Same goes for $n\geq 4$) in our contributions

**Other Strengths And Weaknesses:**

strengths:

-problem considered is initerising

-analysis is non trivial and intuitive

-experiments support validity of the theoretical results

weaknesses:

-results are restricted to monotonic unctions.

-analysis is fairly simple

**Questions For Authors:**

no questions

**Relation To Broader Scientific Literature:**

I am not familiar enough with the broader Scientific Literature in thsi area to give a statement here.

**Theoretical Claims:**

I checked all proofs in the main part and did not have any issues.

---

> ### Author Rebuttal · Authors · 2025-03-26
>
> Dear Reviewer,
>
> Thank you very much for your positive feedback. We appreciate your suggestion and will clarify the definitions of the terms you've highlighted, in our revised manuscript.
>
> If you have any additional recommendations or improvements you'd like to see to elevate your rating further, please let us know—we would greatly appreciate your advice.
>
> Thank you again for your support and constructive comments.

---

### Official Review · Reviewer_gyeZ · 2025-03-17

**Overall Recommendation:** 4

**Summary:**

This paper extends the body of work on monotonic neural networks.
It focuses on relaxing the existing constraints that limits the architecture, use, and performance of such networks.
Specifically, this work identifies limitations within the existing architecture including the use of threshold neurons, which limits the choice of activation functions and the expressivity of monotonic neural networks.
It also identifies potential benefits in reducing the required number of hidden layers for universal approximation with monotonic networks.

In addressing these limitations and others, this work presents theorems that carefully answer the research questions raised.
It proves that universal approximation theorem for non-threshold activation, thereby relaxing the constraint on the choice of activation functions for monotonic networks.
This work carefully reviews the non-negative constraint on the weights and claims that using a non-positive weight constraint is more expressive. It also presented a new parametrization method to better determine the weight constraint.

This work is a valuable contribution to the body of work on monotonic neural representation and its extension to the design of interpretable models.

**Claims And Evidence:**

The claims and evidence are clear and convincing.

The presentation is clear and easy to follow.
The paper provides sufficient background information to understand this work.
It carefully presents and discusses the focus of this work under subsection 3.1, including a clear description of the limitations of existing monotonic MLPs.
Subsequently, the paper provides convincing evidence to support the central claim of addressing these limitations.
Subsections 3.2 provides the proofs that address the use of non-threshold activations and the required number of hidden layers for universal approximation.
Subsection 3.3 provides the justification for a new weight parametrization method.
Section 4 presents the new method for the weight parametrization.

**Essential References Not Discussed:**

None

**Experimental Designs Or Analyses:**

Yes, I checked the experimental analysis.

The experimental results are relevant and illustrate the benefit of the new approach proposed in this work.

**Methods And Evaluation Criteria:**

The methods and evaluation criteria are relevant to the work in this paper.

**Other Comments Or Suggestions:**

Here are some suggestions on typos:

Line 426-426, Column 2, Section 6, Page 8
"We then use this theoretical analysis to construct of a novel parametrization that relaxes the weight constraint making the network less sensible to intialization."
I think "construct of a" should be "construct a". I think "sensible" should be "sensitive". I think "intialization" should be "initialization".

Line 055-056, Column 1, Section 2, Page 2
"Hard monotonicity instead gives guarantees by construction by imposing constraints in the model architecture"
This sentence is not clear to me. Please review.

I do not think the body / text of this paper references Figure 1 directly.

**Other Strengths And Weaknesses:**

ORIGINALITY: This work is original because it clearly identifies the limitations of existing monotonic neural architectures and seeks to address the identified issues.

SIGNIFICANCE: The potential impact of this work in designing easy-to-interpret models underscores its significance.

CLARITY: This paper is well-organized, and its presentation is very clear.


WEAKNESSES:
(1) You should consider additional experimental information to further strengthen the central claim of this work. (Check my comment under Questions For Authors)
(2) The definitions of some notations are not clear. Define the notations under the proofs. (Check my comment under Questions For Authors)

**Questions For Authors:**

I think there are other experiments or discussions that could further strengthen the contributions and central claims of this paper.

(1).  How do non-threshold and threshold activations compare on the training datasets?
(2).  Can you illustrate the effect of the non-negative and non-positive weight constraints on the training dataset?
(3).  What is the additional computational cost of the new weight parametrization method?


Please consider defining the hyperplanes $A_{i_2/i_1}^{+}$, $A_{i_2/i_1}^{-}$ under the Proof for Theorem 3.5 mathematically on pages 5 and 13 (Just like we have in Lemma 3.6). This will improve the readability of the proofs. Also consider adding the explicit definition of $A_{j/i}^{-}$ in equation (6).

**Relation To Broader Scientific Literature:**

The results in this paper are important to the broader scientific literature because they offer a new perspective on the design of monotonic neural architectures. This in turn contributes to the development of explainable models.

**Theoretical Claims:**

Yes, I checked the proofs for Theorem 3.5 and the Lemmas.

---

> ### Author Rebuttal · Authors · 2025-03-26
>
> Dear Reviewer,
>
> Thank you for your constructive review and valuable comments. Below we address each of your concerns in detail:
>
> - _Typos & Figure 1_:
> Thank you for highlighting these issues. We have thoroughly revised the manuscript, corrected all identified typos, and will explicitly cite Figure 1 in the main text.
>
> - _"How do non-threshold and threshold activations compare on the training datasets?"_:
> We omitted this comparison since it has already been explored in [1] (under the name of "Non-Neg-DNN"), where threshold activations were demonstrated to outperform non-threshold approaches. Our proposed method, in turn, matches or surpasses the results obtained by [1]. In the revised version, given that the allowed space is enough, we will add such result in Table 1.
>
> - _"Can you illustrate the effect of the non-negative and non-positive weight constraints on the training dataset?"_:
> We appreciate your question, but we would kindly ask for clarification on this point to ensure we accurately address your concern.
>
> - _"What is the additional computational cost of the new weight parameterization method?"_:
> This aspect is addressed at the end of Section 4.1 (around line 403). Practically, our proposed method introduces negligible overhead and is often computationally cheaper compared to sigmoid-constrained monotonic MLPs, as it avoids the computational costs associated with sigmoid activations and their gradients.
>
> - _"Please consider defining the hyperplanes... Also consider adding the explicit definition of..."_:
> Thank you for noting these ambiguous definitions. We will clearly define these terms in the revised manuscript to ensure precision and readability.
>
> Your feedback significantly helps in improving our manuscript. ICML guidelines this year permit uploading revised manuscripts only during the second half of the reviewing process. We will therefore submit our revised paper as soon as the submission system allows. We hope our clarifications fully address your concerns and further strengthen your positive assessment. We hope these clarifications address your concerns and reinforce your positive assessment.
>
> Thank you once again for your support and valuable suggestions.
>
> [1] Runje, Davor, and Sharath M. Shankaranarayana. "Constrained monotonic neural networks." International Conference on Machine Learning. PMLR, 2023.

---

### Official Review · Reviewer_RAdq · 2025-03-20

**Overall Recommendation:** 3

**Summary:**

This paper proposes a novel Monotonic Neural Network as a universal approximator for monotone functions. Unlike previous works, this approach provides theoretical proof that the proposed Monotonic Neural Networks can serve as universal approximators and successfully removes the constraint of activation function boundedness. As a result, the proposed method enables ReLU-like activation functions to construct monotone networks with universal approximation properties.

**Claims And Evidence:**

The motivation behind studying Monotone MLPs is not clearly stated. The paper lacks a compelling explanation of the practical importance of this research, especially in the current popularity of LLMs and modern deep-learning frameworks. Emphasizing the relevance and potential applications of Monotone MLPs would significantly enhance the paper’s impact. Without this context, the work risks being perceived as a theoretical exercise with limited practical utility.

**Essential References Not Discussed:**

N/A

**Experimental Designs Or Analyses:**

The experimental evaluation is relatively small in scale. Although the authors compare their method with prior works that also use small-scale experiments, the diversity of datasets and task types is insufficient. To convincingly demonstrate the effectiveness and robustness of the proposed method, additional experiments on more complex and diverse tasks are recommended.

**Methods And Evaluation Criteria:**

seems good.

**Other Comments Or Suggestions:**

Based on the above observations, the paper should provide a more rigorous theorem that formally establishes the universal approximation property of the proposed Monotonic Neural Networks under the parameterization described in Equation (12). The current discussion is somewhat informal and lacks the necessary theoretical depth to convincingly prove this property.

**Other Strengths And Weaknesses:**

None

**Questions For Authors:**

None

**Relation To Broader Scientific Literature:**

N/A

**Theoretical Claims:**

The parameterization method described in Equation (12) effectively addresses the weight constraint and eliminates the need to alternate between activations and their point-reflected counterparts manually. However, this parameterization appears to alter the behavior and capacity of the resulting MLP significantly. Consider the following analysis:
- Suppose $x$ is given, and $W_1, W_2$  are drawn from the standard Gaussian distribution. For a standard MLP, $f(x) = Relu(W_1x)$, By linearity of expectation, we have:
$$
\mathbb{E}[f(x)] = ||x|||/\sqrt{2\pi}
$$
For a two-layer MLP: $f(x) = Relu(W_2 \cdot Relu(W_1x))$, It follows that:
$$
\mathbb{E}[f(x)] = \frac{||x|||\sqrt{d}}{2\sqrt{\pi}}
$$
where $d$ denotes the hidden layer dimension.

- Now consider the proposed parameterization: $f(x) = W^+_1 \cdot Relu(y) +  W^-_1 \cdot Relu(-y)，y = W^+_1 \cdot Relu(x) +  W^-_1 \cdot Relu(-x)$ Then:
$$
\mathbb{E}[y]  = \frac{\sum_i{x_i}}{\sqrt{2\pi}}, \quad \mathbb{E}[f(x)] = \frac{d \cdot \sum_i{x_i}}{{2\pi}}.
$$

From this, two key issues arise:
   1.  For a standard MLP, the network can inherently capture non-zero mean outputs. However, in your proposed MLP, if the input is zero-mean normalized, the network may fail to learn meaningful features in terms of $\mathbb{E}[f(x)] $.
   2. Even if we assume that the summation and the norm are equivalent in some intuitive sense, the proposed MLP expands the mean value magnitude by $O(m)$ from one layer to the next, whereas a standard MLP only scales by $O{\sqrt{m}}$. This faster growth suggests that the proposed Monotonic MLP may suffer from instability and increased training difficulty.

In light of these concerns, I strongly recommend further discussion on how the proposed parameterization does not unintentionally introduce new challenges in model stability and learning effectiveness despite addressing gradient vanishing issues in some sense.

---

> ### Author Rebuttal · Authors · 2025-03-26
>
> Dear Reviewer,
>
> Thank you for your insightful review and constructive suggestions. Below, we address each of your points individually:
>
>  - _"Emphasizing the relevance and potential applications of Monotone MLPs would significantly enhance the paper’s impact."_:
>  We fully agree with your suggestion. Monotonic neural networks indeed have diverse applications beyond interpretability and fairness, such as their established use in quantile regression [1]. We will expand the paper to expand on the practical applications of Monotonic NNs, thereby better emphasizing the broader relevance of the topic.
>
>  - _"I strongly recommend further discussion on how the proposed parameterization does not unintentionally introduce new challenges in model stability and learning effectiveness."_:
>  Your point is well-taken and insightful. We will include additional analyses in the appendix examining this issue more closely. However, a thorough and detailed analysis of initialization methods would merit an entire standalone study, thus we will also note it in the future works. We also note this potential concern is common among state-of-the-art architectures employing constrained weights and ReLU activations [2, 3]. (for more details see next point)
>
>
>  - _"However, in your proposed MLP, if the input is zero-mean normalized, the network may fail to learn..."_ & _"This faster growth suggests that the proposed Monotonic MLP may suffer from instability and increased training difficulty."_:
>  We appreciate this important observation.
>  The practical implementation provided in our paper employs PyTorch’s default initialization (Uniform Kaiming or Xe), which does not strictly follow the analysis you provided. Nonetheless, similar results to your one can be obtained on this variant.
>  In the revised appendix, we will add further details addressing your concern, also considering biases, and evaluating empirically our settings to normal MLPs. To anticipate the results that will be reported in the paper, we have analyzed the empirical expansion factor, and even though it can be seen that it's not the same as a normal MLP (as predicted by your analysis), it remains in a reasonable range.
>
>  - _"Additional experiments on more complex and diverse tasks are recommended."_:
>  As mentioned in the Introduction (around line 92), our primary aim is theoretical exploration rather than proposing a novel state-of-the-art architecture. While our theoretical results open opportunities for creating more expressive monotonic MLPs, we believe that the included benchmark comparisons are sufficient for validating our theoretical contributions. Nonetheless, we acknowledge the value of further experiments and agree they would enhance validation.
>
>  - _"The current discussion is somewhat informal and lacks the necessary theoretical depth to convincingly prove this property."_:
>  We agree that the discussion in Section 4.1 (around line 373) could benefit from a more rigorous presentation. We will carefully revise this section, within the constraints of available space, to ensure greater theoretical clarity and rigor.
>
> ICML guidelines this year allow authors to submit revised manuscripts only during the second half of the review process. Accordingly, we will upload the revised version as soon as the submission system permits, with the requested improvements and analysis.
> Your feedback is highly valuable and has helped us improve the manuscript significantly. We hope these clarifications address your concerns and reinforce your positive assessment.
>
> Thank you once again for your constructive comments and support.
>
> References:
> [1] Chilinski, Pawel, and Ricardo Silva. "Neural likelihoods via cumulative distribution functions." Conference on Uncertainty in Artificial Intelligence. PMLR, 2020.
> [2] Runje, Davor, and Sharath M. Shankaranarayana. "Constrained monotonic neural networks." International Conference on Machine Learning. PMLR, 2023.
> [3] Kim, Hyunho, and Jong-Seok Lee. "Scalable monotonic neural networks." The Twelfth International Conference on Learning Representations. 2024.

---

### Official Review · Reviewer_4GJQ · 2025-03-22

**Overall Recommendation:** 3

**Summary:**

This paper constructs universal approximators for monotonic functions with MLPs with non-negative weight constraint and activations that saturate on alternating sides. Based on the result, the paper shows MLPs with convex monotone activations and non-positive constrained weights can also be universal approximators. Furthermore, the authors proposes the pre-activation and post-activation formulations for monotonic ReLU and ReLU' MLPs to address the optimization challenge of monotonic neural networks constructed with weight constraints.

## update after rebuttal
My concerns are well addressed in the rebuttal. I have updated my score to 3.

**Claims And Evidence:**

In Section 4, the paper claims the pre-activation and post-activation formulations can address the weight constraint challenge. However, it seems that the two formulations are not equivalent to the original monotonic ReLU MLP (line 344). Furthermore, to guarantee monotonity and apply Theorem 3.5, the weight constraints still seem to be required.

**Essential References Not Discussed:**

No.

**Experimental Designs Or Analyses:**

Yes.

**Methods And Evaluation Criteria:**

Yes.

**Other Comments Or Suggestions:**

No.

**Other Strengths And Weaknesses:**

Strength:
The paper constructs universal approximators for monotonic functions with one-side saturating activations by simply modifying non-negativity to non-positivity, which is concise and insightful.

Weakness:
The universal approximation result seems direct, considering the construction of Heavyside function (Figure 2) and the universal approximation results of NNs in [1]. The advantages of the paper's results and proof techniques need to be highlighted.

[1] Hornik, Kurt, Maxwell Stinchcombe, and Halbert White. "Multilayer feedforward networks are universal approximators." Neural networks 2.5 (1989): 359-366.

**Questions For Authors:**

1. Can you clarify why the pre-activation and post-activation formulations are equivalent to the original monotonic ReLU MLP and address the weight constraint challenge? For example, give a concrete algorithm of optimizing the monotonic ReLU MLP with the proposed formulations.
2. Can you clarity the advantages of the paper's results and proof techniques? (see Weakness)

**Relation To Broader Scientific Literature:**

N/A

**Theoretical Claims:**

The proofs seem correct.

---

> ### Author Rebuttal · Authors · 2025-03-26
>
> Dear Reviewer,
>
> Thank you for your thoughtful comments and suggestions. Below, we address each of your points in detail:
>
>  - _"The two formulations are not equivalent to the original monotonic ReLU MLP."_:
>    It is unclear whether the "original monotonic ReLU MLP" refers specifically to the setting described in Theorem 1 or to the "naive" ReLU with non-negative weights. For the former case, at the end of page 7 (around line 381), we illustrate how the proposed formulation can be mapped back to the setup used in Theorem 1, establishing that the MLP in Theorem 1 is indeed a special instance of the general formulation we propose. Another reviewer also noted that our description of the former formulation lacked rigor and clarity. We acknowledge this concern, and we plan to significantly revise this section to enhance its clarity and precision in the updated manuscript.
>    In the latter case, if all weights are constrained to be non-negative, both formulations reduce equivalently to a "naive" ReLU MLP. However, this "naive" MLP lacks universal approximation capabilities, highlighting the generality and advantage of our pre- and post-activation formulation.
>
>  - _"The weight constraints still seem to be required."_:
>  In Section 4.1, we demonstrate that by explicitly splitting the weights into positive and negative parts, the need for explicit weight constraints is effectively removed. This is crucial to simplify the implementation, avoiding the activation alternation, and relaxing the need for explicit weight constraints.
>
>  - _"Provide a concrete algorithm for optimizing the monotonic ReLU MLP with the proposed formulations."_:
>  The proposed parametrization can straightforwardly be optimized using standard gradient-based optimization algorithms, consistent with typical neural network training. This is explicitly demonstrated in the provided reproducible code accompanying our submission.
>
>  - _Weakness 1 and "the advantages of the paper's results and proof techniques."_:
>   Existing proof techniques from [1] cannot directly provide a straightforward proof for constrained monotonic MLPs, primarily because they require negative weights in the proof construction$^*$. To overcome this limitation, we provided in Appendix A.1 an alternative, albeit naive, proof built upon the approach from Runje & Shankaranarayana (2023), in a similar spirit as the one you proposed. Although this alternative requires up to 8 layers (a loose bound), while our primary contribution (Theorem 1) significantly improves this by establishing that only 4 layers are necessary.
>
> We sincerely appreciate your valuable feedback and hope the clarifications above resolve your concerns. We believe these refinements strengthen our manuscript and illustrate its novelty and practical value. We appreciate your thoughtful feedback and hope our clarifications have sufficiently addressed your concerns, potentially improving your assessment.
>
> Thank you once again for your time and constructive suggestions.
>
> [1] Hornik, Kurt, Maxwell Stinchcombe, and Halbert White. "Multilayer feedforward networks are universal approximators." Neural networks 2.5 (1989): 359-366.
>
>
> $^*$ from [1]: "By adding, subtracting and scaling a finite number of affinely shifted versions...". They take the difference of threshold functions to create Rect/Box functions. However, we cannot use the "subtract" part since it requires negative weights, while Theorem 1 necessitates non-negative weights.

---

> > ### Comment · Reviewer_4GJQ · 2025-04-02
> >
> > Thank you for your response, which addresses part of my concerns. I can now understand the paper's theoretical contribution of the universal approximation result.
> >
> > However, I am still unclear about the pre-activation and post-activation formulations addressing the weight constraint challenge.
> >
> > - While $f(x) = \text{ReLU}(|W| x + b)$ is always nonnegative, (11) are (12) are not. For example, in (11), let $W=-1, b=0$ and $x\to-\infty$; in (12), let $W=1, x=1$ and $b\to-\infty$. The two formulations are not equivalent to $f(x) = \text{ReLU}(|W| x + b)$.
> >
> > - While these alternative formulations remove the need for explicit weight constraints, they involve rearranging the signs of the weight matrices. Since the motivation for avoiding explicit constraints is to ease optimization, it is unclear whether this rearrangement introduces new optimization challenges. Could you clarify whether this reparameterization changes the optimization landscape, and if so, how?
> >
> > I would be willing to raise my score if these two issues are satisfactorily addressed.

---

> > > ### Author Response · Authors · 2025-04-06
> > >
> > > Dear reviewer
> > > Sorry for the late reply; we thought that in the second part of the review process, we would be able to upload a revised version of the paper that we prepared to address your concerns. However, it is allowed to upload a revised version of the paper only in the camera-ready session, only to accepted papers.
> > > For this reason, we will include directly in this response a summary of the sections we revised in the paper, addressing your concerns. Let us briefly recap the main point of the following response:
> > >
> > >   1. We do not want to be equivalent to $\text{ReLU}(|W|x+b)$. We think this misunderstanding comes from the phrasing used in section 4.1, which will be the section that will be revised and that we will post at the end of this reply. $\text{ReLU}(|W|x+b)$ is not a universal approximator; thus, being equivalent would imply that also our approach is not to be universal approximator. Instead, it was just a way to introduce the reasoning behind the activation switch.
> > >
> > >   2. We don't need to rearrange the sign of the matrixes explicitly, instead we only need to split the positive and negative part of the matrix $W$. Furthermore, consider that in order to apply Theorem 3.5, you would need to use activations with alternating saturation sides. This might require further carefulness in the implementation, and possibly additional hyperparameter tuning. Furthermore, as highlighted by another reviewer, there might be concerns about the initialization. We have also prepared another section of the paper addressing this point, where we show that the proposed method, though it does not scale as a normal MLP, is much better than the naive weight constraint.
> > >
> > >
> > > ## **Summary of the revised Section 4.1** (addressing point 1):
> > > Instead of constraining weights $f(x)={\sigma(|W| x + b)}$, we can separate ${W}$ into its positive and negative parts ${W^+= \max(W,0)}$ and ${W^-=\min(W,0)}$. This allows us to express the affine transformation as
> > > $$
> > > |W|x + b = W^+ x - W^- x + b.
> > > $$
> > > Applying the non-linearity to each term of the equation above individually instead of applying it to $|W|x$, and sharing the bias term, leads to the parametrization:
> > > $$
> > > \hat{f}(x) = \sigma(W^+ x + b) - \sigma(W^- x + b).
> > > $$
> > >
> > > **Proposition:**
> > > Any function representable using an affine transformation with non-negative weights followed by either $\sigma$ or $\sigma'$ can also be represented using the equation above, up to a constant factor.
> > >
> > > **Proof:**
> > > If $W$ has the same sign, one of the two terms in the equation above collapses to ${\pm\sigma(b)}$. Specifically, when $W\ge 0$, the expression reduces to ${\sigma(|W| x + b)-\sigma(b)}$, while when $W\le0$ it reduces to ${\sigma(b)-\sigma(-|W| x + b)}$ instead. To conclude the proof recall that ${-\sigma(-x) = \sigma'(x)}$.
> > >
> > > The additional constant factor can be accounted for in the bias term of the following layer. Therefore, the proposition above covers both cases employed Theorem 3.5, using $\sigma$ and $\sigma'$ as left saturating and right saturating activations.
> > >
> > > This shows that an MLP obtained by stacking at least $4$ blocks parametrized as above is a universal approximator for monotonic functions. Hence, the proposed formulation is more expressive than a simple weight constraint, given that they are only a special case of the equation above.
> > >
> > > A similar reasoning can be applied to the alternative formulation working backward from the last layer of the network, leading to:
> > > $$
> > > \hat{f}(x) = W^+ \sigma(x) + W^- \sigma(-x) + b.
> > > $$
> > >
> > >
> > > ## **Summary of the new appendix chapter** (addressing point 2):
> > > From an empirical perspective, in [Figure](https://ibb.co/C5Z4gTPZ), we can observe the distribution of an output of a multilayer MLP with different parametrization. Unconstrained refers to a naive MLP, Constrained Naive refers to an MLP with $|W|$ as weight parameterization, while pre/post-activation refer to the proposed formulations. While a normal MLP always have 0-mean in predicted value, the switch activation has a very slow tendency to increase the expected output from random initialization, as predicted by theory. However, such slight increase, is notably largely smaller than the one induced by naively constraining the weights to be positive.
> > >
> > > Overall, it can be seen that the activation switch alleviates such behavior by a large factor, thus helping optimization and, thus, performance. All of this while not employing no specific initialization scheme.
> > >
> > > Furthermore, evidence of such improved initialization can be seen in Figures 8, 9, and 10 of the uploaded paper, where we show how a naive weight constraint tends to have an exploding gradient with deeper MLPs using ReLU while vanishing using sigmoid.
> > >
> > > We feel that initialization might be a very fundamental and interesting path to follow. However, we also feel that it deserves a whole work dedicated to it. Therefore, we will include such opportunity in the future works, leaving it as an opportunity for future research.

---

### Decision · Program_Chairs · 2025-05-01

**Decision:**

Accept (poster)

**Comment:**

This paper makes an important theoretical contribution by generalizing previous results on neural network approximation capabilities. Specifically, it demonstrates that MLPs with non-negative weight constraints and properly designed saturation activations can serve as universal approximators for monotonic functions.

The theoretical analysis seems correct and provides valuable insights into the representational power of constrained neural architectures. Now all reviewers found the results to be technically sound and conceptually interesting. I recommend acceptance.